# Enhancing haptic continuity in virtual reality using a continuity reinforcement skeleton

Xinyuan Wang [1], Zhiqiang Meng [1] & Chang Qing Chen [1,2] ✉

Haptic displays are crucial for facilitating an immersive experience within virtual reality. However, when displaying continuous movements of contact, such as stroking and exploration, pixel-based haptic devices suffer from losing haptic information between pixels, leading to discontinuity. The trade-off between the travel distance of haptic elements and their pixel size in thin wearable devices hinders solutions that solely rely on increasing pixel density. Here we introduce a continuity reinforcement skeleton, which employs physically driven interpolation to enhance haptic information. This design enables the off-plane displacement to move conformally and display haptic information between pixel gaps. Efforts are made to quantify haptic display quality using geometric, mechanical, and psychological criteria. The development and integration of one-dimensional, two-dimensional, and curved haptic devices with virtual reality systems highlight the impact of the continuity reinforcement skeleton on haptic display, showcasing its potential for improving haptic experience.

Virtual reality (VR) is expected to spark advances in education, healthcare, and entertainment by enabling immersive interactions[1–4]. In VR, the role of haptic devices is critical, especially given the frequent virtual contact between users and the virtual environment[5–13]. When vision and haptic feedback are not harmonized, it can affect the immersive experience[14] and even produce VR motion sickness[15]. Haptic devices can be generally categorized into interactive object[16–23], kinesthetic devices[24–26], and cutaneous devices[5,27]. Kinesthetic devices, including haptic gloves[24–26] and wearable devices for limbs[28–30], provide force feedback to joints. Cutaneous devices, on the other hand, can generate sensations such as pressure, sliding, and vibration[5,27]. The landscape of cutaneous technology has diversified significantly, with various devices based on different principles, including deformation-driven[18,28,31–35], ultrasonically-driven[36,37], and electrotactile[38,39].

Most haptic devices rely on a matrix of pixels to display pressure, with each pixel moving independently to create tactile sensations[19,22,40–42]. To be thinner and lighter, devices that are fluidically actuated[43], shape memory material actuated[44,45], or electrostatically actuated[31,46] have been designed. However, pixel-based devices encounter limitations when depicting moving contacts, such as stroking and exploring the shape of objects, as shown in Fig. 1a. In these cases, haptic information between

pixels is lost, leading to a sensation of discontinuous motion marked by the alternating rise and fall of pixel points. As shown in the upper part of Fig. 1b, the blue portion represents the target shape to be displayed, while the red dashed line highlights the peaks that are not effectively rendered. Addressing this issue by merely reducing pixel size could lead to a more complex system[47] and a reduction in actuation travel. Firstly, decreasing the pixel pitch by a factor of $k$ will increase the number of pixels per unit area by a factor of $k^2$. According to the analysis by Biswas and Visell[13], a skin of the size $1\,cm \times cm$ should comprise at least $10{,}000 \times 10{,}000$ pixels to fully restore the haptic, which is far greater than the currently obtainable pixel density. Secondly, for many two dimensional (2D) thin wearable haptic devices, when a constant off-surface displacement is required, if cell size is reduced, the strain will rise sharply. For example, to generate a 2 mm off-surface displacement, the strains on the pixel surface are about 40% and 300% when the cell size is 4 mm and 1 mm, respectively (See Supplementary Note 1). Due to the limitations of the maximum elongation rate of the material, there is a trade-off between the cell size and the actuation travel distance.

To address these challenges without increasing pixel density, one promising strategy is to add connections between pixels and interpolate the regions lacking haptic feedback. Notable studies

[1]Department of Engineering Mechanics, CNMM and AML, Tsinghua University, Beijing, P.R. China. [2]Mechano-X Institute, Tsinghua University, Beijing, PR China. ✉e-mail: chencq@tsinghua.edu.cn

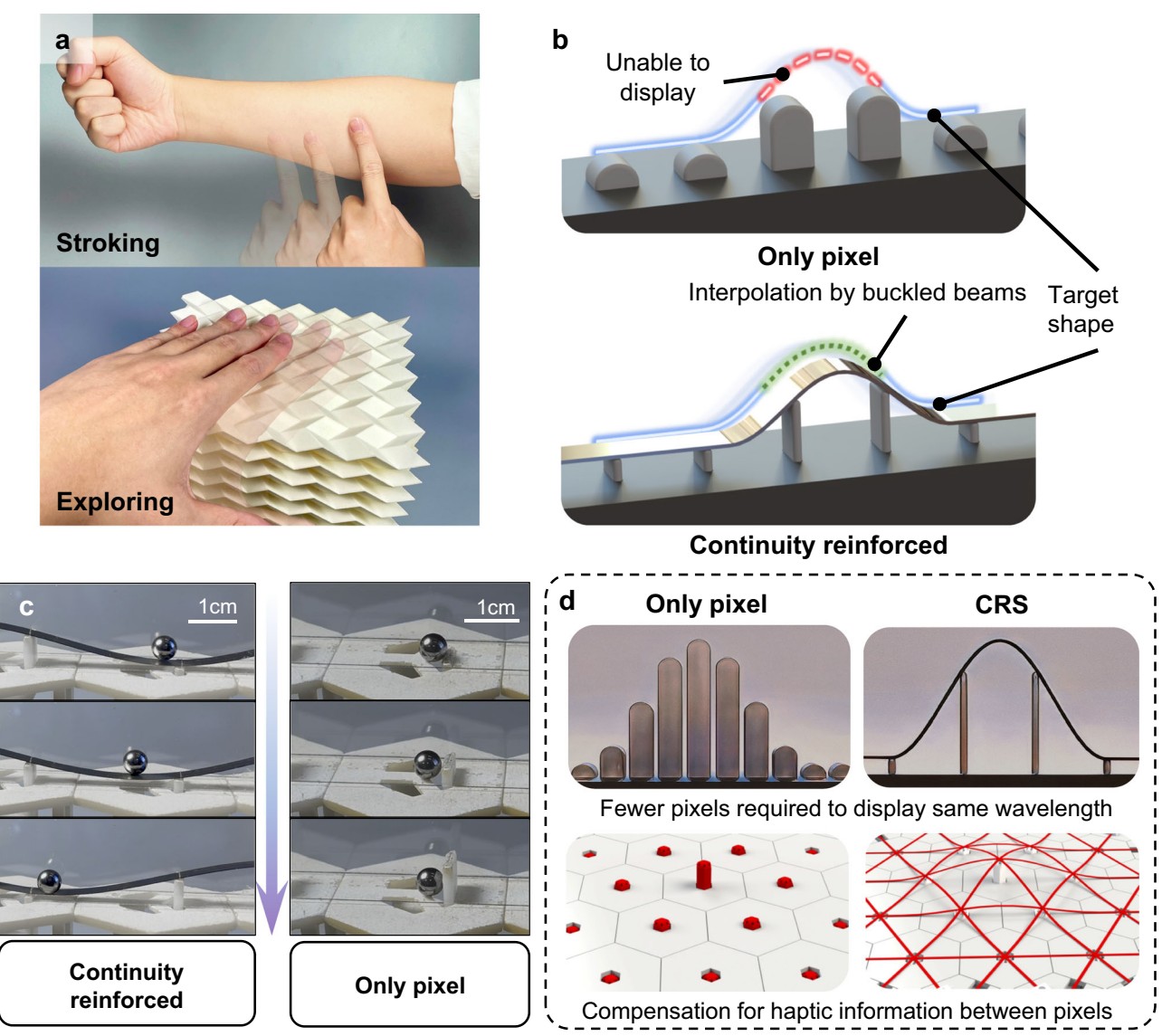

**Fig. 1 | Continuity reinforcement in haptic display. a** Examples of continuous contact in VR. **b** Concept of continuity reinforcement. **c** The process of controlling a small ball with or without the Continuity Reinforcement Skeleton (CRS). **d** Advantages of the CRS.

addressing this issue include forming surfaces with origami[34,48], kirigami[49], or semi-solid surfaces[50], offsetting the unit[31], developing surfaces with negative Poisson's ratios[51], and manipulating magnetic fluids[52]. However, most existing haptic devices cannot conformally display the maximum off-plane displacement in an area beyond the pixel point, due to the loss of haptic information or the linear connection between pixels. Establishing a structurally simple, easily miniaturized, and robust method without increasing the pixel density for continuity reinforcement in haptic devices remains a significant challenge.

Here, we introduce a continuity reinforcement skeleton (CRS) for haptic displays by leveraging the inherent smoothness of beams under bending and buckling. As shown in the lower panel of Fig. 1b, the CRS interpolates the haptic information based on the pixels' off-plane displacement marked by a green dashed line. This mechanism is further exemplified in Fig. 1c, which shows that a CRS device can control the movement of a ball with a diameter of 5 mm across a pixel pitch of 30 mm. Using the CRS, continuously adjustable off-plane displacements of pixels can be transformed into laterally continuous motions that can stabilize the ball at arbitrary positions between pixels (see Supplementary Movie 1). The CRS

design can reduce the required pixel density to display the shape of the same wavelength, as shown in Fig. 1d. It can also deliver haptic stimuli on the lines connecting the nearest neighboring pixels in 2D haptic devices. With its simple structure, the CRS can be easily miniaturized and integrated with various actuators. We will show that the CRS can also significantly reduce position distortion and shape distortion.

## Results

### One dimensional (1D) CRS device and quantification of continuity

The 1D CRS device, shown in Fig. 2a, features a design with five pixels that can be independently adjusted in height, each powered by a small linear servo. To address the loss of haptic information typically experienced between pixels, the 1D CRS device (Fig. 2a) incorporates a 0.1-mm-thick steel beam on the pixels. Each pixel point is connected to the beam using a fishing line, facilitating movement along the beam's length while managing its off-plane displacement. The choice of fishing line as a connector serves a dual purpose: it reduces interference with the haptic interface while ensuring adequate connection strength. Furthermore, the device uses the intrinsic interpolation nature of buckled beams. This

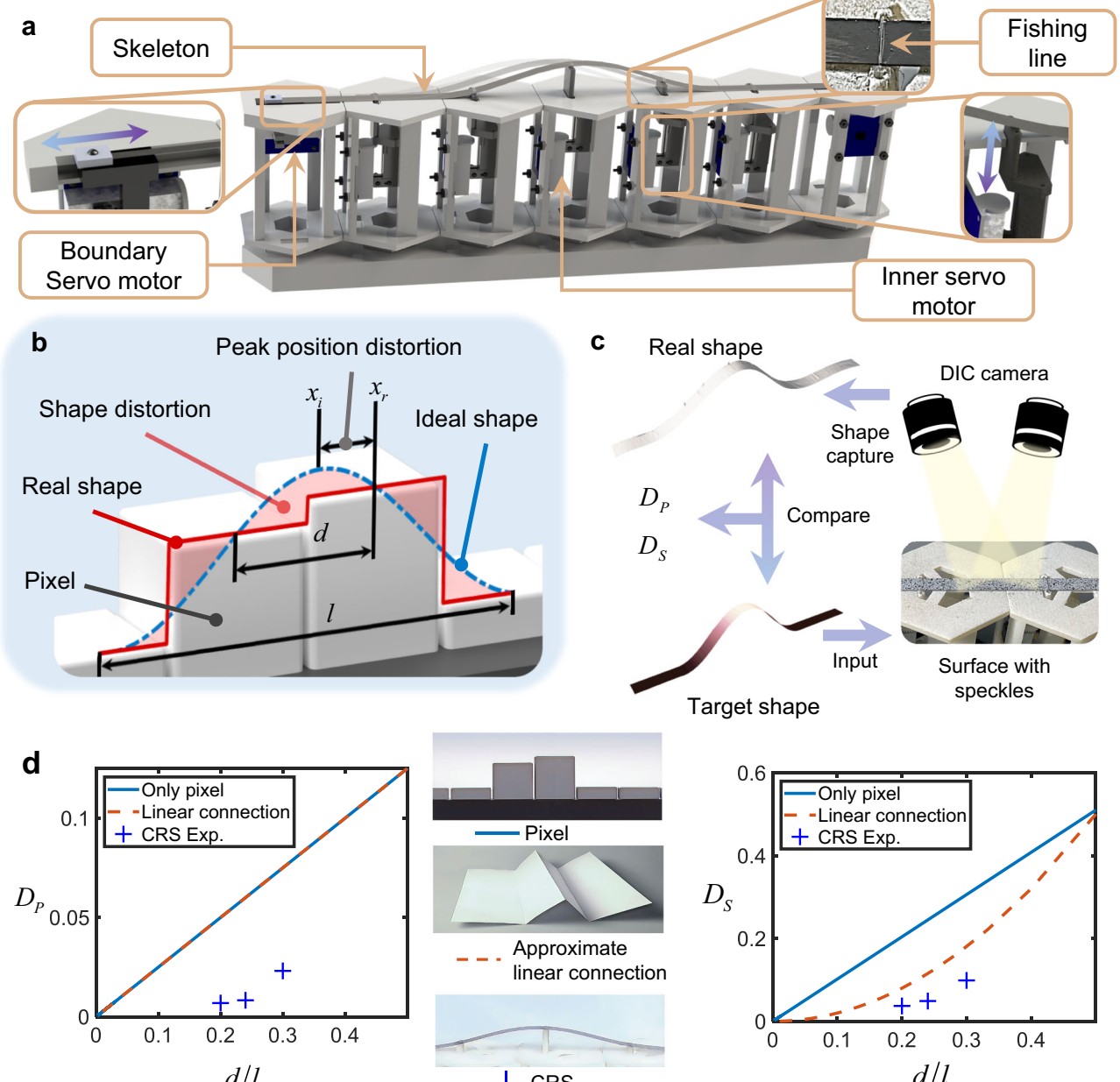

**Fig. 2 | Quantification of continuity. a** An 1D Continuity Reinforcement Skeleton (CRS) device. **b** Definition of distortion in haptic displays. **c** Measurement of distortion of a haptic display using Digital Image Correlation (DIC) Method. **d** Three basic configurations of haptic displays' different $D_P$ and $D_S$ curves.

is achieved by compressing the beam at its two ends, an action managed by boundary servos. These boundary servos control the total lengths of the beam in the display area to satisfy the geometric constraint (see Supplementary Movie 2 for the display effect).

To assess the effect of the CRS, it is necessary to define metrics to quantify continuity, as illustrated in Fig. 2b, where a column of pixel points with a spacing of $d$ display a single-peaked sinusoidal wave with a wavelength of $l$. The blue dashed line represents an ideal waveform, while the red solid line represents the actual shape. Two distinct types of distortion is evident when the spacing is comparable to the wavelength: shape distortion and peak position distortion. Shape distortion refers to the discrepancy between the ideal and the actual waveforms, while the peak position distortion is defined as the deviation between the ideal peak position and its actual location. To quantify these distortions, we assume the positions of ideal peaks $X_i$ ~ $U(x_1, x_n)$, in which $x_1$ and $x_n$ represent the first and last possible peak

positions. A mean peak position distortion $D_P$ can be defined as:

$$D_P = E\left(\frac{|x_r - X_i|}{l}\right) \tag{1}$$

in which $x_r$ is the actual peak position deviated from the ideal position $X_0$. It should be emphasized that $E(|x_r - X_i|/l)$ is the expectation algorithm in probability, not the Young's modulus. The mean shape distortion $D_S$ can be defined as:

$$D_S = E\left(\sqrt{\frac{\int_{X_i-l/2}^{X_i+l/2}(\varphi(x,X_i) - \psi(x,X_i))^2 \mathrm{d}x}{\int_{X_i-l/2}^{X_i+l/2}(\varphi(x,X_i))^2 \mathrm{d}x}}\right) \tag{2}$$

in which $\varphi(x,X_i)$ is the ideal shape and $\psi(x,X_i)$ is the actual shape (See Supplementary Note 2 for the expression of $\varphi(x,X_i)$).

The experimental measurement of $D_P$ and $D_S$ is illustrated in Fig. 2c. First, speckles are sprayed on the surfaces to be measured. Then, the off-surface displacement is measured using the Digital Image Correlation (DIC) method. The measured waveforms are compared with the corresponding input waveforms to obtain the distortions, which are then averaged to get the experimental mean distortions (see Supplementary Note 3 for the processing of the experimental data).

The two distortions are mainly determined by the ratio of pixel distance and displayed wavelength $d/l$. Clearly, the smaller the pixel distance, the less the distortion for a fixed wavelength. From the perspective of interpixel connection, 1D haptic display devices can be categorized into three types, as shown in the middle panel of Fig. 2d. Besides the pixel display and the CRS, origami-inspired displays can also deliver haptic information and can be approximated as connecting pixel points with straight lines[48].

The shape and position distortion curves are shown in the left and right panels of Fig. 2d, respectively. The blue solid line and the orange dashed line represent the theoretical distortion curves for pixel display and linear connection. For both the pixel-only and linear connections, there exists a theoretical formula $D_P = 0.25d/l$. Because linear connection cannot display peaks between two pixels, thus cannot reduce $D_P$. For the pixel-only condition, there are empirical formulas for shape distortion $D_S = 0.994d/l$; for the linear connection, one has $D_S = 1.545(d/l)^2$ (see Supplementary Note 2 for the derivation of the mentioned formulae). The shape distortion $D_S$ of linear connection has a quadratic dependence upon $d/l$ and is lower than the pixel distortion curve, implying that the linear connection has lower shape distortion for the same pixel density. The blue crosses in the two graphs are the experimentally measured distortion of the CRS device. Both $D_P$ and $D_S$ of the CRS device are lower than pixel-only and linear connection devices. In particular, $D_P$ is an order of magnitude lower than the two types of haptic devices.

## Mechanical model of CRS

The beam in the CRS is expected to be soft to reduce the load on the actuators while strong enough to support the skin. Therefore, it is necessary to construct a mechanical model for the CRS to determine the thickness of the beam, as shown in Fig. 3a for a CRS on an elastic foundation with $d/l = 3$. The elastomer as a substitution of skin is simplified to a Winkler foundation[53], i.e., it is assumed that the distributed load on the beam is proportional to its deflection with a scaling factor of $\beta$. The peak is located at the midpoint of the two pixels (see Supplementary Note 3 on the analysis of this situation). Because the ends of the beam have a zero-degree angle to the base and are compressed along the length direction, it can be simplified to moveable ends without rotation. The axial load of the beam is $N$ and the load at the support point is $P$. The deflection of the beam $y(x)$ has the following trigonometric series expression[53]:

$$y(x) = 4Pl^3 \sum_{n=1,2,3\ldots}^{\infty} \frac{\left[1 - \cos\frac{n\pi(l-d)}{l}\right]\left(1 - \cos\frac{2n\pi x}{l}\right)}{n^4 16\pi^4 EI - 4n^2\pi^2 l^2 N + 3\beta l^4} \quad (3)$$

where $l$ is the wavelength and equal to $3d$, $E$ is the Young's modulus of the beam, $I$ is the beam's moment of inertia, and $n$ represents the wavenumber of symmetric buckling mode. The axial critical loads for each order of buckling can be obtained by making the denominator zero, i.e., $N_{cr}^{(n)} = (n^4 16\pi^4 EI + 3\beta l^4)/4n^2\pi^2 l^2$. When $n=3$, the wavelength of the buckling mode is equal to the pixel spacing, causing the shape of the beam to be unconstrained by the pixels and collapsing of the beam, as shown in the upper panel of Fig. 3b. To suppress the 3rd order mode, it is necessary to ensure $N_{cr}^{(1)} < N_{cr}^{(3)}$. Thus, the no-collapse

condition can be obtained as:

$$\Delta = \frac{16\pi^4 EI}{27\beta d^4} > 1 \quad (4)$$

When $\Delta > 1$, $N_{cr}^{(1)} < N_{cr}^{(3)}$ and no collapse occurs. On the contrary, when $\Delta < 1$, the beam collapses when pressed into the elastomer. Equation (4) is consistent with the general intuition that the stronger the beam, the softer the skin, and the smaller the pixel spacing, the less likely it is to collapse.

In Fig. 3b, $\Delta$ in the upper and lower panels are 0.2 and 1.7, respectively, showing the collapse and no-collapse scenarios. The corresponding force-displacement curves during loading are shown in Fig. 3c, and the sudden jump of the curve when collapse occurs is evident. Furthermore, the no-collapse condition can be written as $\Delta = (16\pi^4/27)(E/\beta)(I/d^4)$, where the dimensionless parameters $E/\beta$ and $I/d^4$ represent material and geometric features, respectively. With the parameters of geometry and material as horizontal and vertical coordinates, we can obtain the phase diagram of collapse, shown in Fig. 3d. The blue and red parts represent the no-collapse and collapse regions separated by a black dashed line $\Delta = 1$. Triangles and circles mark the experimental results of collapse and no-collapse, consistent with the phase diagram prediction.

Using the above no-collapse condition, a device can be designed to verify the continuity reinforcement effect of the CRS by interacting with elastomer, shown in Fig. 3e. The upper surface of the elastomer is covered with speckles to facilitate displacement measurement using the DIC method. Figure 3f illustrates the silicone elastomer's different displacement fields in the $Y$-direction without or with the CRS. The deformation concentrates around the pixels without the CRS. In contrast, the deformation field of the elastomer with the CRS varies continuously (see Supplementary Movie 3).

In addition to meeting the no-collapse condition discussed above while designing the beam in CRS, a significant change in the beam length due to the axial compression is not desired for control requirement. Note that when $\Delta \gg 1$, $\Delta l = N_{cr}^{(1)}L/EA \approx \pi^2 b^2 L/3l^2$ is material-independent, with $L$ and $b$ being the length and thickness of the beam, respectively. Therefore, thinner beams with larger Young's moduli should be selected to satisfy the no-collapse condition and the negligible length change condition. A more detailed discussion of the mechanical behavior of the CRS is provided in Supplementary Note 5.

Besides, it should be noted that the CRS is subject to additional constraints compared to pixel haptic displays. First, the maximum curvature of the CRS is constrained by the yield strength of the material, which limits the range of pixel movement. Second, since the CRS requires boundary servos compression, its curve length increase needs to be less than the boundary servos' travel distance. For a quantitative discussion of the limitations of the CRS on the range of pixel movement, please see Supplementary Note 8.

## 2D CDS device

A 2D CRS device is shown in Fig. 4a. The device comprises 19 hexagonal internal pixels that can be independently controlled and 18 servos on the boundaries that compress the ends of the beams. In the top right inset of Fig. 4a, three beams are stacked on each other and tied to the pixels with fishing lines. To measure its off-surface displacement, a layer of spandex fabric with speckles covers the CRS and is sewn to the moving part of the pixel with fishing line. Figure 4b illustrates how the CRS device displays a non-developable surface that cannot be obtained from a plane without tearing, creasing, and stretching. Due to the frame structure of the CRS, the beams only undergo bending deformation, which prevents the generation of large internal stress and enhances deformation capacity. Figure 4c exhibits the shape with and without the CRS when displaying a target waveform located at the midpoint of two pixels. The result with the

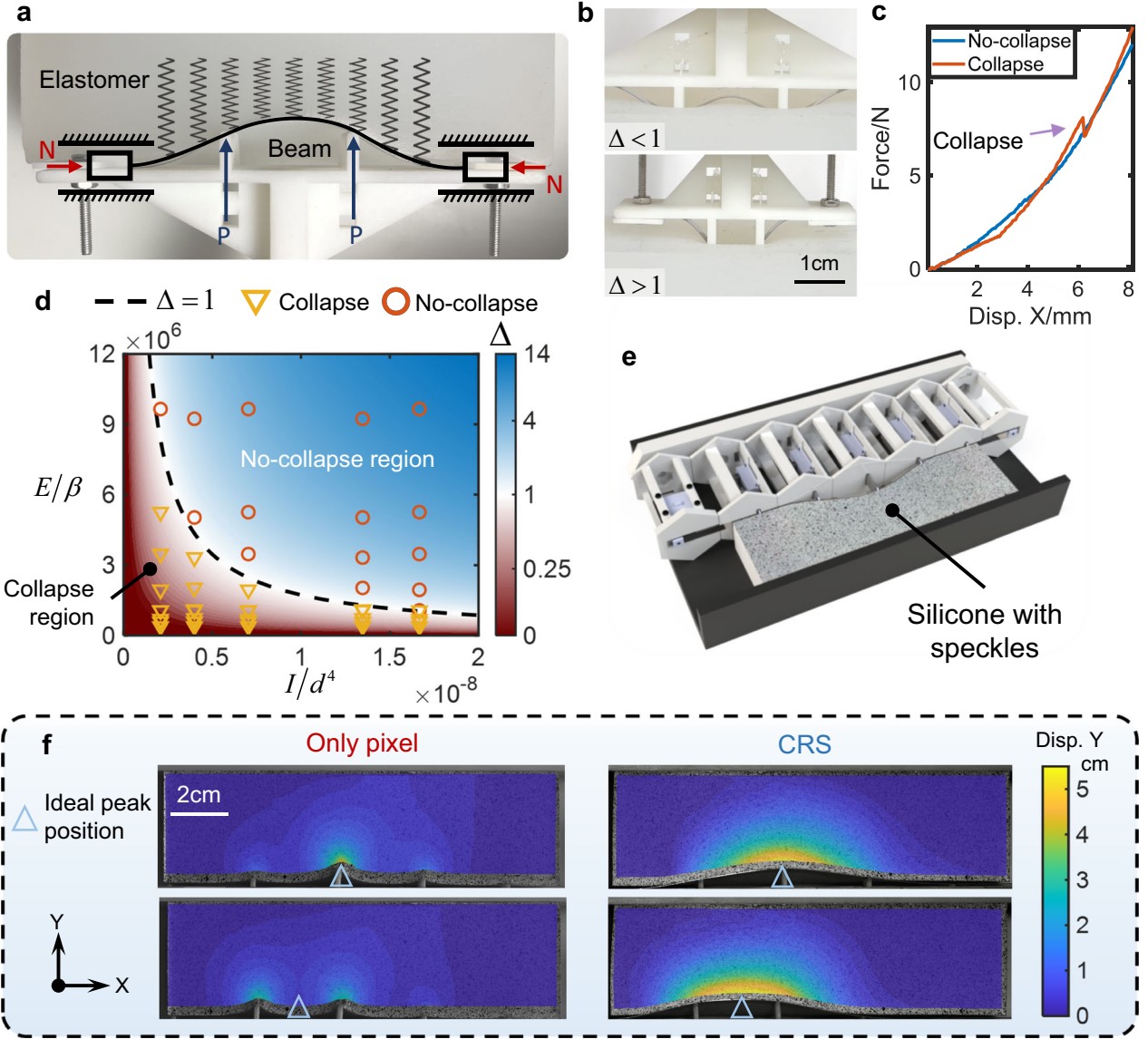

**Fig. 3 | Mechanics analysis of Continuity Reinforcement Skeleton (CRS). a** The mechanical model of the CRS contacting with an elastomer. **b** Schematic of collapse vs. no-collapse. **c** Force-displacement curves during loading with and without collapse. **d** Phase diagram of collapsing, where the triangle and circle markers represent experiment result. **e** A device to demonstrate the continuity reinforcement effect using silicone as a substitute for skin. **f** CRS's effect on the *Y*-direction displacement field in silicone.

CRS is more restorative to the target shape, while the surface without the CRS forms two small bumps responding to the two pixels points. The shape in Fig. 4c has been exaggerated 10 times in the *z*-direction to show their differences.

Figure 4d and e show the effect of displaying ripple spreading and movement of a circular point. The top half is the actual image, while the bottom half shows the target shape and the displacement field measured by the DIC method. Comparing the displacement fields shows that the surface formed by the CRS experiences continuous motion instead of sequential rise and fall of the pixels (see Supplementary Movie 4).

To quantify the reinforcement effect, it is necessary to extend the concept of haptic distortion to 2D cases. Let the 2D point contact be a rotated sinusoidal surface shown in the target shape of Fig. 4c and use it as the benchmark to illustrate display quality. The equation of the target shape can be found in Supplementary Note 2, and $(X_i, Y_i)$ is the peak position of the contact point which is uniformly and randomly distributed in the display region. The peak position distortion can be

obtained as

$$D_P = E\left(\frac{\sqrt{(x_r - X_i)^2 + (y_r - Y_i)^2}}{l}\right) \qquad (5)$$

where $x_r$ and $y_r$ are the coordinates of the actual peak position and are functions of $(X_i, Y_i)$. For square pixels, $D_P = 0.3825d/l$; for hexagon pixels, $D_P = 0.3510d/l$. It is also possible to define the shape distortion in 2D as

$$D_S = E\left(\sqrt{\frac{\iint_S (\varphi(x,y,X_i,Y_i) - \psi(x,y,X_i,Y_i))^2 \, \mathrm{d}x\mathrm{d}y}{\iint_S (\varphi(x,y,X_i,Y_i))^2 \, \mathrm{d}x\mathrm{d}y}}\right) \qquad (6)$$

in which $\psi(x,y,X_i,Y_i)$ is the actual shape, $S$ is the range covered by the ideal shape. Square pixels have an empirical formula $D_S = 1.412d/l$. For hexagon pixels, $D_S = 1.318d/l$ (see Supplementary Note 2).

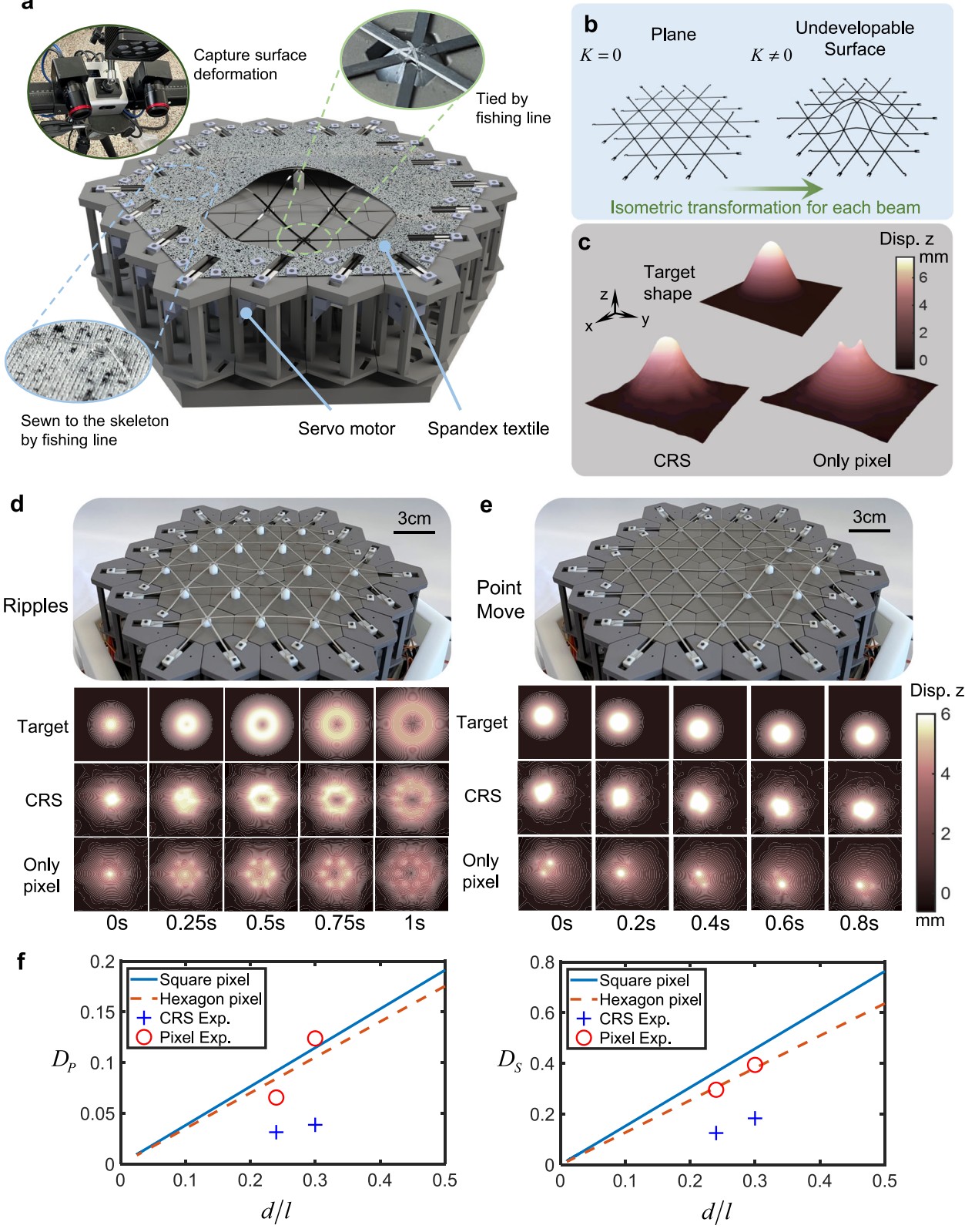

**Fig. 4 | 2D Continuity Reinforcement Skeleton (CRS) device. a** Structure of 2D CRS device. **b** The 2D surfaces constructed by a grid of beams while generating a non-developable curvature. **c** Comparison of the shapes with and without the CRS when showing a specific target shape. **d** Display of the spread of ripples and displacement field of $z$ from Digital Image Correlation (DIC) method. **e** Display of the movement of points and displacement field of $z$ from DIC method. **f** Theoretical curves of distortion for two types of pixels displays and the experimentally measured distortion.

Based on the theoretical and experimental results, we can obtain the distortion curves, see Fig. 4f, g. The blue solid lines and the orange dashed lines represent the theoretical distortion curves for square and hexagonal pixel points, respectively, while the blue crosses and the red circles represent the experimentally measured distortion with and without the CRS (see Supplementary Note 3 for experiment data processing). It is clear that the CRS can reduce the shape and position distortions by half.

## Visual-haptic integrated virtual reality with CRS

Haptic displays combined with visual VR can create immersive experiences as shown in Fig. 5a. Based on current VR technology, movements such as touch are made more vivid by integrating haptic displays with the CRS. The two characters pictured are wearing headsets and haptic devices to communicate in a VR environment. As one person's fingertips touch the other one's hand, the haptic display device will restore continuous touching for the user. This scenario demonstrates the future of remote immersive communication, allowing people to meet, talk, and have physical contact with their family regardless of the distance by integrating visual, auditory, and haptic senses.

Figure 5b shows a demonstration of visual-haptic fusion interaction. In this virtual environment, when moving a finger within the cyan hexagonal area near the palm of the character, the 2D CRS device displays a bump that follows the fingertip in real time and controls its amplitude by the depth of the finger's pressure. When the finger writes the number "9", the 2D CRS device also displays the track of writing the number "9". The green cross in the figure indicates the approximate location of the peak on the haptic display, which is recorded every 0.5 s (See Supplementary Movie 5). As shown in the experiment given in Supplementary Note 9, the CRS device, when compared to a pixel-haptic, can increase the accuracy of digit identification from 69.9% to 84.7%. In order to explore the display effect of the CRS under extreme cases, e.g., when the size of the digit is close to that of the pixel pitch, the adopted average height and width of the digit in this test are only about 0.63- and 0.41-pixel pitch, respectively.

The framework of the visual-haptic integrated system is shown in Fig. 5c. The VR headset is connected to the computer via USB streaming and displays the VR environment. The finger movement is captured by the controller and passed to MATLAB via User Datagram Protocol (UDP) communication. The computer generates the shape of the haptic surface using MATLAB and sends commands to the servo controllers through serial communication, controlling the motion of each servo via Pulse Width Modulation waves.

In future applications, the CRS may have the ability to combine with other types of actuators and be overlaid on the body as a wearable device. However, as shown in Fig. 5d, most of the human body surfaces are curved. Therefore, it is essential to build curved CRS devices in Fig. 5e to verify whether the CRS system can work properly on a curved surface. The basic structure of this device is the same as that of the 1D CRS, except for its curved surface. As illustrated in Fig. 5f, the beam can also maintain its shape during motion, reflecting high continuity and conformability (see Supplementary Movie 6).

## Discussion

A CRS with physically driven interpolation of haptic information has been developed. The CRS can enhance the continuity of shape movement on planar or curved surfaces without increasing pixel density. The key to continuity reinforcement is to utilize pixels to guide the buckling of the beam. Thanks to the beam's interpolation nature, this design allows wave peaks to be displayed statically or dynamically at arbitrary locations between the pixel gaps. The CRS has the potential to combine with various existing pixel-based haptic displays by incorporating beams on the contact interface. The simple structure of the CRS allows for fabrication in various sizes and easy miniaturization. Moreover, the no-collapse condition remains intact when the skeleton is proportionally scaled-down. When combined with a haptic device with higher pixel density, the CRS can depict more detailed and sharper shapes. The CRS also expands the haptic feedback area from dots to a network woven by skeletons. The beam's inherent smooth bending deformation ensures the safe and comfortable interaction.

An attempt has been made to quantify the continuity of haptic displays, using the mean position and shape distortion during the movement of sine waves or rotating sinusoidal surfaces with different wavelengths. This evaluation criterion of continuity is chosen for several reasons. The first rationale for this criterion is that the contact with this shape is common in daily life, such as fingertips, nibs, and balls. Secondly, many complex waveforms can be expanded as a sum of several sine functions of different wavelengths. The distortion theory can be used to estimate the display effect of the sharpest edges in complex graphics (See Supplementary Note 4). Lastly, this evaluation criterion reveals the average display quality of the pattern when it spans the entire domain, as pressure peaks may not always align precisely with the pixel point and must be able to move. For a haptic device capable of displaying sharp edges, high continuity means that it can simulate the sensation of a blade scraping against the skin, while low continuity means that those sharp edges cannot move conformally.

In this paper, the discussion on the CRS systems focuses on the display of pressure. In order to display complete haptic information, further work can focus on displaying higher pixel densities, tangential force, vibration, and temperature. Although the CRS can improve the digit identification rate, pixel density still needs to be increased to display more detailed tactile information to further improve performance. Then, the beams in the CRS can move along their length direction, generating tangential forces on the skin with friction. Regarding vibration, there is also a need to investigate the mechanics of wave propagation in beams to display textural features. The CRS, which is made of metal, has good thermal conductivity and is also expected to integrate temperature displays. The CRS is promising not only for multimode haptic display and VR technology, but also for mechanical information processing[54–57].

## Methods

### CRS device

The CRS device in this paper is constructed using photosensitive resin through 3D printing. The beams used in the CRS were laser-cut from SUS301 stainless steel, specifically 12Cr17Ni7, in accordance with GB/T 20878-2007. The Young's modulus of this steel is 193 GPa. Due to structural design requirements, the thickness and width of the steel beams vary in each drawing. The CRS units in Fig. 2 use steel beam that is 4 mm wide and 0.1 mm thick. Meanwhile, the beam in Fig. 3 is 4 mm wide and 0.15 mm thick. To ensure structural strength and prevent interference, the beam width of the 2D CRS device is reduced to 1 mm and the thickness is 0.15 mm.

The driving devices consist of linear servos made by AFRC, with its weight of 1.5 g, and dimensions of $21.2 \times 15 \times 6$ in mm. The maximum travel of the servo is 9 mm and the maximum torque of the motor is $0.24 \, \mathrm{kg \cdot cm}$. Operating on a 6 V input voltage, this motor moves at 0.08 s/cm, i.e., it takes 0.072 s to move from the lowest to the highest point. In order to control multiple servos in a harmonized manner, a 24-channel LSC-24-V2.3 servo controller manufactured by Hiwonder was used in this study. Powered by an 8 V DC regulated power supply, this controller is controlled by a PC via TTL serial communication. The surface can be precisely controlled to display the desired shape using a MATLAB program, which converts the haptic information into control signals and calculates the length of compression needed on each side of each beam to achieve the target shape (See Supplementary Note 7

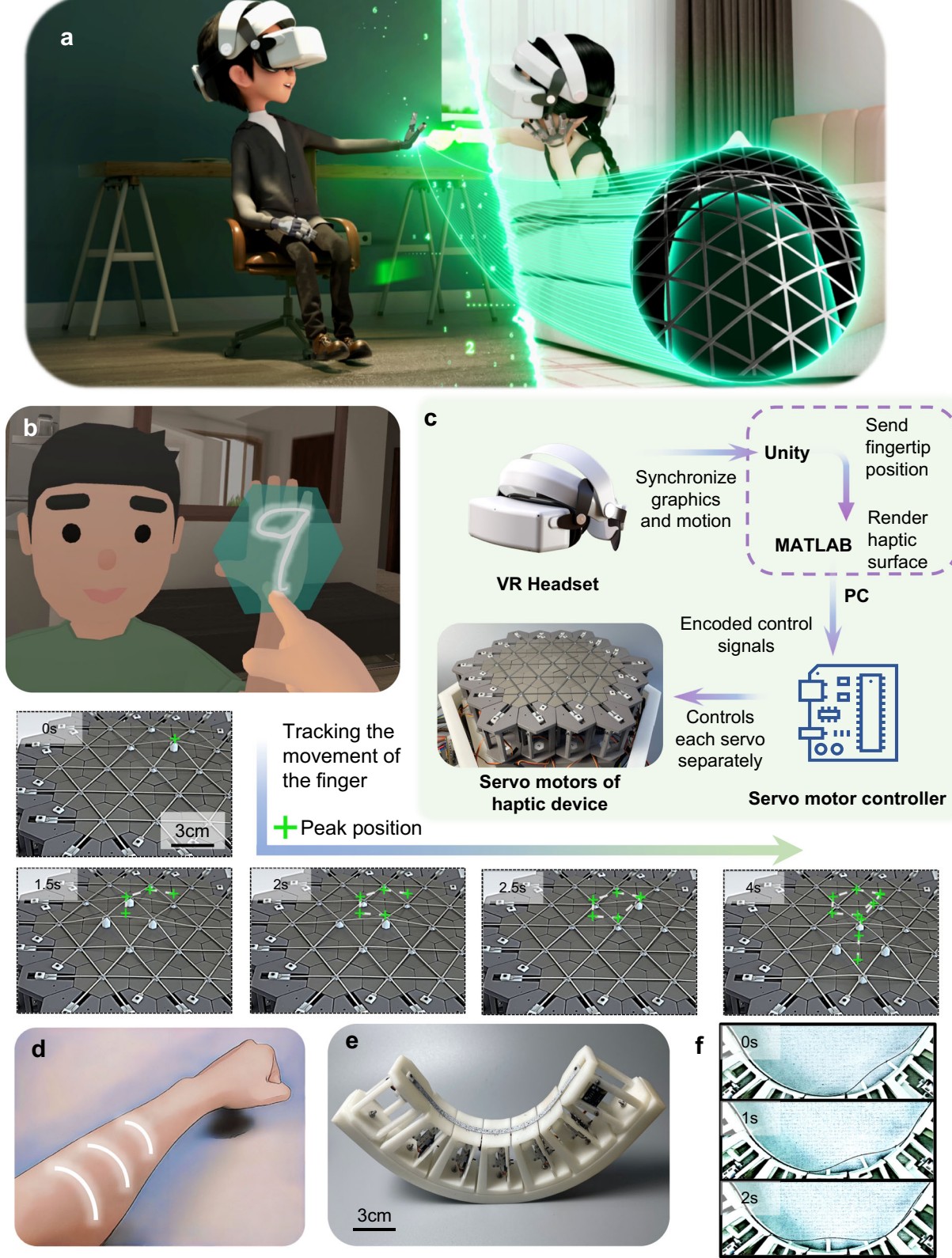

**Fig. 5 | Haptic-visual integrated virtual reality with Continuity Reinforcement Skeleton (CRS). a** Schematic of communicating remotely via VR headsets and haptic display devices. **b** Linkage of the 2D CRS device to the user's finger while writing on the character's hand. **c** Haptic-visual combined system diagram. **d** Curved human body surface. **e** Curved CRS device. **f** Demonstration of the movement process of curved CRS devices.

for control algorithm). Given the 2D CRS device consists of 19 pixels and 18 boundary servos, exceeding the control capacity of a single controller, the servos are controlled by two controllers connected to the same PC.

To ensure that the beams can be controlled by the pixel points for off-plane displacement, while allowing the beams to move in the along-length direction, the beams were bound to the pixels with high-strength fishing line of 0.148 mm diameter. In order to measure the

effect of the CRS on 2D surfaces, the CRS was covered by a layer of sprayed speckle spandex fabric with high elongation, soft and smooth texture. The spandex fabric is likewise sewn to each pixel point with fishing line, which provides restraint and has a low coefficient of friction with steel. Due to their small diameter, fishing line does not significantly affect DIC measurements and haptic display (see Supplementary Note 6 for the assembling of CRS units).

## VR environment

The VR haptic interaction scene in Fig. 5b was written using the Unity platform. The scene is simulated on a personal computer and displayed in the VR headset via USB streaming. The VR headset used here is the Oculus Quest 2 by Meta. The scenario provides a virtual environment within a home in which the user can walk freely and control the position of their hands using VR controllers. When the grip button on the back of the controller is pressed, the character extends its index finger and captures the position of the finger pointer in real time. When the fingertip approaches the cyan area that represents recognition in Fig. 5b, the VR program sends the position of the finger relative to the center of the CRS recognition area $(x_f, y_f, z_f)$ to MATLAB via UDP communication for conversion into haptic signals, where the $z_f$ direction is perpendicular to the recognized region. The shape of the display of the 2D CRS is determined by the equation

$$z = \frac{1}{2} z_f \left[ 1 + \cos\left( 2\pi \sqrt{(x - x_f)^2 + (y - y_f)^2} / 90 \right) \right] \qquad (7)$$

where $\sqrt{(x - x_f)^2 + (y - y_f)^2} \leq 45$. The haptic rendering thus takes the shape of a rotating sinusoidal surface with a diameter of 90 mm whose height $z_f$ is determined by the depth of the press and position $(x_f, y_f)$ is determined by the fingertip's position. The compression control algorithm for the boundary servos is calculated directly from the equation of the surface, referring to the Supplementary Note 7. By continuously stroking in the recognition area, the movement of the finger can be displayed continuously on the 2D CRS, and the height of the bump can be controlled by the depth of the press in the recognition area. The latency of the CRS device is 75 ms. When the CRS device works with a VR environment, the latency of the whole system is about 160 ms (See Supplementary Note 11 for details on the measurement method).

## Mechanical and DIC test

In the paper, the mechanical properties of the CRS were analyzed using silicone instead of human skin. Three pieces of silicone with different hardness were used: the softest was Ecoflex 00−10, the hardest was Ecoflex 00−30, and the medium hardness was Ecoflex 00−30 and thinner in a ratio of 2:1 by mass. The coefficient of subgrade reaction $\beta$ of the three types of silicon was obtained by uniaxial compression testing (See Supplementary Note 5). In the compression experiments, a press head with a sinusoidal shape was used for the test to get as close as possible to the force applied to the CRS when it comes into contact with the skin. The force-displacement curves obtained from the tests are shown in the Supplementary Note 5. The testing machine was an Instron 5900 equipped with a 5 kN force transducer.

In order to quantify the effect of the CRS, the DIC method is utilized to measure the 3D displacement, with the XTDIC-CONST-ST 3D full-field strain measurement and analysis system. All the DIC results have been calibrated by a calibration plate to ensure the authenticity of the data. The absolute displacement measurement accuracy is about 0.01 pixel.

## Reporting summary

Further information on research design is available in the Nature Portfolio Reporting Summary linked to this article.

## Data availability

All technical details for producing the figures are enclosed in Methods and Supplementary Information. Source data are provided with this paper.

## Code availability

The codes that support the findings of this study are available from the corresponding author Chang Qing Chen upon request.

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

## Acknowledgements

This work is supported by the National Natural Science Foundation of China (Nos. 12132007, and T2488101).

## Author contributions

X.W., Z.M., and C.Q.C. developed the concept. X.W. designed the devices, control system and software, developed the theory, and finished the experiments. X.W., Z.M., and C.Q.C. wrote the manuscript.

## Competing interests

X.W. and C.Q.C. are inventors of a patent filed by Tsinghua University (202410256252.4, granted on 23 January 2025) related to this work. Z.M. declares no competing interests.

## Ethical approval

This study was conducted in compliance with the relevant ethical regulations. The Tsinghua University Science and Technology Ethics Committee (Medicine) approved the study, and the participants provided written informed consent.
