## [Transparent Peer Review file · Nature Communications]

Enhancing Haptic Continuity in Virtual Reality using a Continuity Reinforcement Skeleton

Corresponding Author: Professor Changqing Chen

Version 0:

Reviewer comments:

Reviewer #1

(Remarks to the Author)

The authors propose a haptic display highlighting the continuity of the actuating surface in 1D & 2D. The authors tried to fill the continuity gap of the flexible surface actuated by the vertical pins by employing physical interpolation of bending and buckling of the flexible beam. Although it is good to see the authors tackle the typical problem of pin-based haptic display, I still have concerns as below.

Performance as a haptic display

Although the authors mentioned this is a haptic display, there is no evaluation or validation for the physical or perception of the haptic display performance. Without these validations, the proposed works would fit more as a deformable surface. I want the authors to fully validate the exerted pressures given by the proposed work as well as the same concept with different material properties of the surface (e.g., when using different Young's modulus for actuating material). This will provide a clear picture of the physical performance of the haptic display. Moreover, it would be interesting to look into the perception performance of the continuous surface for the user. (or need a good reference on the human perception limitation on perceiving the continuity of the surface) This will determine whether the proposed continuity-reinforced structure affects user perception of a haptic display.

Replicability & Actuator Placement

Although the suggested equation and results validate the performance of the continuity mechanics, they provide little information on how to design the proposed haptic display. As an HCI & haptic researcher, I am curious about a systematic way of configuring the proposed haptic display with my own mechanical design structure. In order to do this, a systematic demonstration using either an algorithm or equation on how to configure actuation coordinates for the given geometry should be provided. In this way, readers could replicate the authors' proposed work with any customized shapes or structures. This would also provide meaningful information on the limitations of the proposed work in terms of supported volume, size, length, etc.

Technical Detail

There are some places in the paper where technical implementation details are missing. For example, the detail of the mechanical structure should be demonstrated for clarity in Figure 2. Although this paper is in nature communication, as an HCI researcher, I find that Figure 5 with VR context misses lots of information. For instance, I would like to find out what method has been applied to reduce the haptic effect with the CRS device. The system diagram would be necessary to explain how the overall system works as a VR haptic system. Moreover, the algorithm block on how the CRS haptic display was rendered would be needed in the 'Methods' section rather than simply explaining how Meta Quest and MATLAB were configured.

More Related Works

There have been various previous works on shape-changing display with flexible covers shown below. I would recommend the authors to include these works and highlight the contribution of the proposed work.

- https://dl.acm.org/doi/pdf/10.1145/3064663.3064677?casa_token=AW2P6YAqqlgAAAAA:6lmyaP6sZohr3_mwO-br4rDIZlorIXUBL9diB8dbxTB_yoBNSPGJTN3dWOqNdB-A65rOkxqrDAzflYY

- https://dl.acm.org/doi/pdf/10.1145/3313831.3376834?casa_token=1O3EeUOMvfUAAAAA:iT8ScGswxndP2aXHgVPo-

tXN7I-85gSN2ur39_KW9ruCNFyn8ZcjhVcAym5TuNY-h4-XfTuhtmbMG48

- <https://www.frontiersin.org/articles/10.3389/frobt.2019.00080/full>

- https://dl.acm.org/doi/pdf/10.1145/1709886.1709928?casa_token=4FQYZ-tsNagAAAAA:vEdYxJ1tFTj3k3z5-oj4A-mEH_UBiAh9df_0W6gfNkYBqVhuhO77zICWh2QZcxJzUGnrf6hOtCQWHFU

With above reasons, I am bit hesitant to accept the paper as is. Also, I think it might require multiple iterations since above suggestions require additional new context that need a new set of review.

Reviewer #2

(Remarks to the Author)

This work presents a continuity reinforcement skeleton (CRS) that can be placed on top of a pin array used to display shape. This skeleton allows for the smoothing of motions, and allows the haptic display to obtain smooth shapes and continual motion of a peak point along a beam. It overcomes the fundamental challenge of pin arrays, that for better approximations of smooth shapes more and more pins are needed. CRS is applied to a row of actuator and a single beam, an array of hexagonal pins, and a curved figure. The hardware presented is a novel contribution, but some additional analysis would further strengthen the contributions of this paper.

The hardware design consists of a set of thin beams that go over the top of the pins, constrained at each pin with a loop of fishing line. Additional servos at each end of the beam control the overall length of the beam. This hardware design is a compelling solution. It is able to render smooth shapes and continuous motion. However, whereas conventional pin arrays cannot create smooth shapes, the pin array with the CRS cannot produce shape with sharp edges, which may be desirable in haptic displays. Adding this CRS enables new abilities, but at the expense of others. There may be cases where rendering the edge of a surface, rendering braille characters, or rendering a point contact are desirable for haptics.

I would also be interested to know more of the limits of the device. How does the CRS affect the maximum height that can be achieved by the peak? There are cases as in figure 3 where the peak of the beam extends beyond the height of the pins. What is the additional force required from the pins to deform the beam? Is there a case where the strain limit in the beam could become the limiting factor?

It is interesting that there is no discussion of human perception in this paper. The authors have instead discussed that they instead use mathematical metrics, but not ones that consider human perception. This is logical considering the focus of the device is on the display, but the contribution would be strengthened if these mechanical marks could be shown to correlate with perceptual metrics. It seems to me that the ability to display sharp edges still be very valuable for some perception tasks.

In figure 3 on the phase diagram, the amount of experimental data does not make the transition clear. Having more data to analyze this phase transition would be a valuable addition

The computational analysis focuses on determining when the beam will buckle. Is a computational model also used to control the position of the boundary servos and heights of the pins. It is unclear to me how the position of the boundary servos are controlled.

The evaluation primarily uses digital image correlation, which capture well the performance of the device. The authors, in both modeling and experiments, consider the beam applying force to the a silicone structure to represent the human body, which is a good approximation.

The videos and figures are well done and very explanatory. If possible, adding a scale reference in the videos would be useful.

Reviewer #3

(Remarks to the Author)

What are the noteworthy results?

This paper proposes and evaluates a method to create 1D and 2D haptic displays augmenting traditional haptic-pixel displays using a Continuity Reinforcement Skeleton (CRS). In a nutshell, the CRS is defined by a series of steel beams that connect haptic-pixels enabling a continuous shape rendering. The beams are also actuated by end effectors to improve the response of the system.

The paper also discusses future works that could extend the CRS approach, by providing additional haptic cues, such as tangential forces or vibrations. These highlights additional benefits over traditional haptic-pixel displays.

Will the work be of significance to the field and related fields? How does it compare to the established literature? If the work is not original, please provide relevant references.

Haptic displays is an active research field which deal with the challenge of proposing effective and simple systems to deliver

rich kinesthetic and tactile feedback. The proposed system provides a modular and cost effective approach to deliver cutaneous feedback during the exploration of virtual surfaces.

However, authors mainly use the system to display simple shapes, it remains unclear how the system could render more complex shapes in which multiple pixels are actuated arbitrarily.

Another potential limitation is that the system is unable to render sharp edges. This could be mitigated by the increase of the density of the pixel display, although reducing the gain of the use of the CRS.

Although the proposed VR scenario is compelling, the added value of the VR seems limited and it remains unclear how the pixel-based display could be easily worn by users.

Finally, one aspect that is not discussed in detail on the paper, is how the gaps in the 2D version of the device are covered, and whether this is actually required. This doubt is raised due to the lack of explanation of the intended use of the device, is the user interacting with it, or it is always in contact with the surface of the skin. From the content of the paper, it seems the latter, but it should be clarified.

Does the work support the conclusions and claims, or is additional evidence needed?

The paper discusses the requirements of the CRS in terms of shape continuity and the mechanical model. The proposed data shows the viability of the system and the forces capable of delivering.

Authors mainly discuss their results against discreet haptic-pixel devices, and in particular, all evaluations are conducted considering the same prototypes, removing the CRS structure. These evaluations, although they ensure that both systems are equivalent in terms of pixel density, are strongly biased due to the large spacing between haptic-pixels. Traditional haptic-pixel displays will exhibit higher density to reduce the gap between pixels. This generates unrealistic configurations for haptic-pixel displays. This somehow generates a biased comparison in which the CRS display is always superior. Comparing the system with higher density displays would have provided a fairer description and authors could have discussed the density increase to achieve comparable results.

Two aspects not discussed in the paper raised my attention. First, authors do not discuss the coupling between the "haptic-pixels" and the beams. Multiple activations of pixels attached to the same beam might generate too much tension.

Second, authors do not discuss the latency of their system. Considering that it is intended for real-time use cases, latency should be assessed.

Are there any flaws in the data analysis, interpretation and conclusions? - Do these prohibit publication or require revision?

The analysis is sound and carefully assesses the continuity of the proposed prototypes evaluating the forces and breaking points of the device. Precisely, two major assessments are provided, the deformation of the connecting beams, and the pressure that the beams can support from external forces (e.g. skin).

Is there enough detail provided in the methods for the work to be reproduced?

Authors provide enough information to replicate the proposed system and conduct the evaluations.

Other comments

The title could be a bit more informative, such as "Continuity Reinforcement Skeleton for Pixel-Based Haptic Displays"

(In 20) Authors talk about "a novel perspective compared to traditional psychological assessment", I do not see the novelty on their assessment.

(In 22) The system has not been tested in VR with real users, so the statement is weak and evidence is missing.

(In 37) This statement is false, a wide range of haptic devices exist to generate pressure and forces (already discussed in the previous paragraph of the paper). Maybe authors wanted to state that "Most haptic devices that rely on matrix of pixels to display pressure,..."

(In 85) In several parts of the paper, authors talk about the "user's skin" it remains unclear the type of interactions that are considering.

(In 88) "it minimizes interference with haptic feedback while ensuring robust connectivity" this sentence is not justified.

(In 93) In several sentences authors talk about "haptic experience" or "user experience", it remains unclear what authors are referring with it. Considering the evaluation conducted authors, it should refer to the quality of the generated shape.

Although the paper is easy to read, it will benefit from a full English proof-read.

References that can be of interest of authors:

Sturdee, Miriam, and Jason Alexander. "Analysis and classification of shape-changing interfaces for design and application-based research." *ACM Computing Surveys (CSUR)* 51.1 (2018): 1-32.

Zhu, L., Jiang, X., Shen, J., Zhang, H., Mo, Y., & Song, A. (2022). TapeTouch: A handheld shape-changing device for haptic display of soft objects. *IEEE Transactions on Visualization and Computer Graphics*, 28(11), 3928-3938.

Wang, Jiabin, et al. "A novel shape-changing haptic table-top display." 2017 International Conference on Optical Instruments and Technology: Optoelectronic Imaging/Spectroscopy and Signal Processing Technology. Vol. 10620. SPIE, 2018.

Boem, Alberto, and Giovanni Maria Troiano. "Non-rigid HCI: A review of deformable interfaces and input." *Proceedings of the 2019 on Designing Interactive Systems Conference*. 2019.

Version 1:

Reviewer comments:

Reviewer #1

(Remarks to the Author)

Hi. I have read the authors' revisions. I found that there is a message "Error! Reference source not found" throughout the revision letter. (I am not sure what this relates to)

Overall, I liked the thorough revision reflected by the authors. Still, I have below comments.

Below are my thoughts on the revised context.

- I liked the additional user study with the proposed system. However, I also felt that the current reported performance is low (<70%). Could we confidently say use this system as claimed with this user performance? I know it is better than the pixel-based approach baseline; the authors should provide a rationale for this relatively low performance and how it can be overcome in future iterations. This should be explicitly mentioned in the Limitation section to prevent any overclaim.
- For supplementary Discussion 7, I am glad that the authors provide a generalized method to expand the form factor of the proposed system. However, I would like the authors to clarify when to use the direct control and interpolation methods with integrated figures if possible.
- For Supplementary Fig.22, what is the role of "Auxiliary node"? I would also like the author to integrate Fig. 21 and Fig. 22. It would be even better if the authors could include the explanation of the direction control algorithm in this figure as well as mentioned earlier.
- The limitation regarding volume, size, and length should be explicitly mentioned in the body text rather than only explained in the supplementary.
- I am worried about the system's current delay, which is well over the typical haptic device requirement (<50~100ms). I think this would be a major limitation in terms of haptic devices, and this should be clearly stated in the body text of the paper. In addition, the authors should suggest how this delay could be reduced in future iterations. In my opinion, the delay is the most critical performance in providing meaningful haptic experience to the user especially when working within AR/VR environment.

Reviewer #2

(Remarks to the Author)

The authors have made substantial improvements to their report, primarily by adding studies on the users perception of the haptic device, as well as in providing details on the control methodology for the device.

I am still unclear about the effect of the pixel force. I understand that the majority of the actuation force comes from the servos at the boundary of the device, but isn't there also the case that for some shapes far from the natural buckled modes of the beam, large forces from the pixels are still required. For example, in the case of a steep peak between two constrained pins, it seems like the neighboring pins would still have high force. I did not understand this statement in that regard: "If the force on the pixels need to further reduce, the control algorithms can be fine-tuned to increase the compression."

Reviewer #3

(Remarks to the Author)

First of all, I would like to thank the authors for their thorough revision and the detailed information provided in the cover letter. Through the revision authors address most of the remarks of the reviewers, and have provided additional details and

discussion in the manuscript and the supplementary materials. I believe that the changes introduced are relevant and help to clarify/discuss topics which were briefly addressed or not addressed. In overall, I'm in favor of accepting the contribution, I've just three major points that should be addressed.

- While authors have provided additional details on the complexity of the shapes that can be displayed with the CRS device, most of the examples mainly discuss low frequency features (few curves). One interesting feature of pixel displays is their capacity of render high-frequency features, which somehow is limited by the addition of the CRS. Authors should discuss advantages and disadvantages of using the CRS device (e.g., increased continuity at the expense of decreased flexibility on the shapes that can be rendered).

- The relevance of the VR environment example provided on the paper is still unclear. In its current state, it is merely an illustration. Authors stated in the cover letter "haptic devices do not need to track users' movements", I do not see how this is specifically linked with the CRS device. To me, it does not bring anything interesting to the paper and could be simply removed. It would have been more interesting to discuss use cases in which the CRS display could be used to overcome the limitations of pixel-based displays.

- Authors provided a measurement of the latency of the system, 440ms for haptic interface is really high. It might be interesting to provide details/discuss why such a high latency and how it could be reduced.

Version 2:

Reviewer comments:

Reviewer #3

(Remarks to the Author)

I would like again to thank the authors for the revised version. At this stage, I do not have any further comments.

Reviewer #4

(Remarks to the Author)

We greatly appreciate the authors' comprehensive response and additional experiments to address all earlier comments and/or concerns. Akin to original reviewers' opinions/comments, this is definitely an interesting and useful design for various VR-HCI systems.

In alignment with Reviewer #1's comments, we note that the additional user study on visual-haptic fusion interaction yielded a somewhat low accuracy rate (<70%). The response letter has addressed this. Although we agree with the authors that the limited number size could be somewhat a contributing factor to this slightly lower accuracy rate, we would like to kindly note that there are alternative aspects in which this experiment could be further enhanced.

- The size of the numbers in the visual and haptic displays differs significantly. From Supplementary Video 5, the number displayed covers the entire area of the virtual hand, which should be approximately 18 cm in height and 8 cm in width for an average adult hand (source: <https://www.healthline.com/health/average-hand-size>). However, the actual average size of the numbers is only 2.43 cm in height and 1.48 cm in width. Further clarification could be beneficial.

- The configuration of the digital identification experiment differs from what is shown in Figure 5, as the average size of the written numbers is significantly smaller. We believe the reason of this modification is worth discussion in the main text.

- Stroke order can be another critical factor influencing identification accuracy. From Supplementary Figure 25, we notice that all 10 numbers are written in a single stroke, which may not be necessarily ideal, particularly for numbers like "4" and "5." Additionally, the starting points for "8" and "9" could be adjusted. For instance, "8" could start from the center, and "9" could begin slightly lower. We believe these factors can appear very interesting for general readers.

Another minor concern/comment is whether the CRS system would reduce the system's load-bearing capacity, such as whether the system will be more likely to collapse when there is a certain downward pressure from the palm on the CRS system? Stability while using this system can be an interesting discussion aspect, for the authors' information.

Reviewer #5

(Remarks to the Author)

Version 3:

Reviewer comments:

Reviewer #4

(Remarks to the Author)

The authors have made significant revisions with extra experimental results to address the concerns listed in the review. We do not have further comments at this moment.

Reviewer #5

(Remarks to the Author)

Responses to reviewers' comments

In the following, the authors' replies, summarized in a one-to-one manner, are shown in **blue**, with the changes made in the revised manuscript denoted in **red**. In addition, while using the original numbering of figures/references in the manuscript (e.g., Fig. 1, ref. 1), we add the prefix "R" for those presented in this response letter (e.g., Fig. R1, ref. R1).

Reviewer #1

General comment: The authors propose a haptic display highlighting the continuity of the actuating surface in 1D & 2D. The authors tried to fill the continuity gap of the flexible surface actuated by the vertical pins by employing physical interpolation of bending and buckling of the flexible beam. Although it is good to see the authors tackle the typical problem of pin-based haptic display, I still have concerns as below.

Authors' reply: We are grateful to the reviewer for his/her invaluable time spent reviewing our paper. The comments are of great help in improving our paper.

Comment 1.1: Although the authors mentioned this is a haptic display, there is no evaluation or validation for the physical or perception of the haptic display performance. Without these validations, the proposed works would fit more as a deformable surface.

Authors' reply: We agree with the reviewer's suggestion and have conducted Human Computer Interaction (HCI) experiments to demonstrate the impact of continuity on haptic display. The details of these HCI experiments are provided in Supplementary Discussion 9. The revised corresponding content in the main text and the Supplementary Information is as follows.

(Page 12 of main text)

...The green cross in the figure indicates the approximate location of the peak on the haptic display, which is recorded every 0.5 seconds (See Supplementary Movie 5). As shown in the experiment given in Supplementary Discussion 9, the CRS device, when compared to a pixel-haptic, can increase the accuracy of digit identification from 50.3% to 67.8%.

(Page 17 of Supplementary Information)

Supplementary Discussion 9 Haptic identification of handwritten digits

To investigate the impact of continuity on the haptic display, we measured and compared the accuracy of digit identification using palm haptic with and without CRS. In the experiment, volunteers were instructed to feel the movement of contact (similar to main text Fig. 5b) and identify the corresponding number of the trajectory, see Supplementary Fig. 25. The two haptic devices share the same shape and a pixel pitch of 30 mm. The shapes of the displayed digits from 0 to 9 are shown in Supplementary Fig. 26. For each number, we collected five handwriting patterns, with five samples of the number 3 depicted in Supplementary Fig. 26. These digits have an average width of 14.8 mm and an average height of 24.3 mm, which are smaller than one pixel pitch to enhance the challenge of identification. The shape of the contact point is determined by Supplementary Equation (8), in which $h = 10\text{mm}$ and $l = 100\text{mm}$.

Supplementary Fig. 25 Pixel haptics and CRS haptic devices.

A total of eight volunteers (seven males and one female) used their right hand for number identification. Each volunteer had 40 learning opportunities, i.e., the numbers were displayed according to the volunteer's request to facilitate their familiarization with the task. The volunteers then performed four sets of identification experiments, with each group recognizing ten random numbers (which could be repeated). Half of the volunteers tested the haptic device with CRS first, then measured pixel haptics, while the other half in reverse order.

The confusion matrices shown in Supplementary Fig. 27 were plotted based on the volunteers' identification of numbers. The numbers in the blocks indicate the frequency of the identification results corresponding to specific ground truth. For instance, the "1" at the lower-left corner of the pixel haptic's confusion matrix means that the volunteers mistakenly answer "0" while displaying "9" one time. Blue blocks represent correct identification, while orange blocks represent incorrect ones. By improving the continuity of the haptic display, the accuracy in identifying numbers with complex strokes (such as numbers 6, 4, and 9) increased. The average accuracy of the CRS device was 67.8%, while that of pixel haptic was 50.3%. By enhancing the motion continuity, the accuracy of user identification improved by about 17.5%.

Supplementary Fig. 26 The trajectories of handwritten digits in the identification experiment.

Supplementary Fig. 27 The confusion matrices of pixel and CRS haptics in digit identification.

Comment 1.2: I want the authors to fully validate the exerted pressures given by the proposed work as well as the same concept with different material properties of the

surface (e.g., when using different Young's modulus for actuating material). This will provide a clear picture of the physical performance of the haptic display.

Authors' reply: It should be noted that accurately measuring the pressure exerted on the skin during an experiment is challenging without affecting the pressure field. We utilized the finite element method (FEM) to simulate the pressing of the skeleton against the skin. The obtained numerical results show that changes in the Young's modulus of the CRS material do not affect the pressure distribution on the skin surface when the no-collapse condition is satisfied,. This is because the pressure on the skin is solely determined by the surface shape, irrespective of the material of the CRS. This results indicate that the performance of the CRS is not sensitive to material properties.

(Page 18 of Supplementary Information)

Supplementary Discussion 10 Pressure distribution on the skin

Note that measuring the pressure on the skin surface is often challenging, due to interference from accurate pressure sensors on the pressure field. However, the pressure distribution can be simulated using the FEM method. In order to calculate the pressure distribution, it is necessary to first measure the mechanical properties of the skin. As an example, we measured the uniaxial compressive stress-strain curve of silicone, as shown in Supplementary Fig. 28. Silicone was chosen to mimic the skin. The obtained Young's modulus was 63.9 kPa and the Poisson's ratio was about 0.25.

Supplementary Fig. 28 Uniaxial compression test of silicone.

Supplementary Fig. 29 Numerically simulated and experimentally measured strain field under the skin. The simulations and measurements were obtained using the FEM and DIC methods, respectively.

In the FEM simulations, the boundary conditions were set to match those in the CRS, with a maximum indentation depth of 3 mm into the skin. Other parameters include a beam width of 4 mm, thickness of 1 mm, and a sinusoidal shape with a wavelength of 51 mm. In order to validate the finite element model, the FEM predicted strain field is compared with the corresponding experimental measurement by the

Digital Image Correlation (DIC) method. The strain fields presented in Supplementary Fig. 29 show good agreement between simulations and experiments. The observed minor difference is attributed to the different locations of the pressure imposed by the two methods: the CRS is applied to the center of the skin in the FEM method, while the indenter is applied to the edges in the DIC method because the DIC method can only measure surface deformation.

The FEM predicted pressure fields on the skin surface, induced by the CRS with different Young's moduli, are shown in Supplementary Fig. 30. The pressure field remains unchanged when the modulus of the beams is increased to 10 and 100 times the original value, which indicates that the pressure field is independent of the material of the beams when the no-collapse condition is satisfied. Consequently, the CRS is displacement-controlled and insensitive to the choice of materials.

Supplementary Fig. 30 Pressure field on the skin generated by CRS.

Comment 1.3: Moreover, it would be interesting to look into the perception performance of the continuous surface for the user. (or need a good reference on the

human perception limitation on perceiving the continuity of the surface) This will determine whether the proposed continuity-reinforced structure affects user perception of a haptic display.

Authors' reply: Agreed. We have analyzed the effect of continuous surfaces on perception performance, see our reply to Comment 1.1. In addition, Biswas and Visell¹ have pointed out that an ideal haptic display that can fully simulate the sense of touch should have 10,000×10,000 free-moving pixel points on a 1 cm ×1 cm skin and be driven at a frequency of 1,000 Hz. However, existing haptic displays do not meet this requirement and as a result, the discontinuities introduced by the pixels are perceivable by the human body. Therefore, there is a need to compensate for the discontinuities during the movement of the contacts by CRS. We have added this discussion in our introduction.

(Page 3 of the main text)

...by a factor of k will increase the number of pixels per unit area by a factor of k^2 . According to the analysis by Biswas and Visell¹, a skin of the size 1 cm ×1 cm should comprise at least 10,000×10,000 pixels to fully restore the haptic, which is far greater than the currently obtainable pixel density. Secondly, for many two dimensional (2D) thin wearable haptic devices...

Comment 1.4: Although the suggested equation and results validate the performance of the continuity mechanics, they provide little information on how to design the proposed haptic display. As an HCI & haptic researcher, I am curious about a systematic way of configuring the proposed haptic display with my own mechanical design structure. In order to do this, a systematic demonstration using either an algorithm or

equation on how to configure actuation coordinates for the given geometry should be provided. In this way, readers could replicate the authors' proposed work with any customized shapes or structures.

Authors' reply: We would like to thank the reviewer for the comment. Detailed control algorithms of the CRS have been included in Supplementary Discussion 7, as shown below. We hope that these algorithms can serve as a reference for future studies.

(Page 11 of Supplementary Information)

Supplementary Discussion 7. Control Algorithms of Continuity Reinforcement Skeleton

To control the CRS display surface, it is necessary to calculate the travel distance of the boundary servos. The key problem is to calculate the compression of each CRS for a given specific pattern. Two methods have been proposed to perform the calculation: the direct control method and the interpolation control method.

Supplementary Discussion 7.1 Direct Control Method

The direct control method involves calculating the travel distance of the boundary servos, based on the surface equations. This method is suitable for surfaces that can be described by a simple analytical equation. In this paper, all the CRS experiments are controlled using this method.

Consider a surface $z = \varphi(x, y)$ with a CRS beam placed on the xOy plane. The projection of the CRS onto the plane is given by $y = y_l(x)$. The start and end points of the display area of the CRS beam are x_s and x_e , respectively. Denoting the respective original and deformed lengths of the CRS beam within the display area by L_0 and L_t , the total travel distance ΔL for the servo motors on this CRS beam is given by

$$\Delta L = L_t - L_0 = \int_{x_s}^{x_e} \sqrt{1 + \left(\frac{dy_l}{dx}\right)^2 + \left(\frac{\partial z}{\partial y_l} \frac{dy_l}{dx} + \frac{\partial z}{\partial x}\right)^2} dx - L_0 \quad (1)$$

It should be noted that the above integral expression does not always have an analytical solution. A more effective approach to calculating the length of a curve is to discretize it as a polyline, as a first-order approximation. Let there be a list of uniformly distributed points on the curve (x_j, y_j, z_j) , in which $j=1,2,\dots,n$. For planar CRS problems, the above list of points can simply be generated at equal intervals in the x -direction. The distance ΔL can then be estimated by

$$\Delta L = L_t - L_0 = \sum_{j=1}^{n-1} \sqrt{(x_{j+1} - x_j)^2 + (y_{j+1} - y_j)^2 + (z_{j+1} - z_j)^2} - L_0 \quad (2)$$

Note that in contrast to the previous formulas in integral form, the above calculation of lengths does not require derivation, and the formula is also applicable to CRS on curved surfaces as long as the discretization is done in a reasonable way.

Supplementary Discussion 7.2 Interpolation Control Method

If the equations of the surface are unknown and only the heights at the pixel points are available, the travel distance of the boundary servos can be approximated using the third-order Hermite interpolation, a method for segmental interpolation using cubic functions while ensuring second order continuity at the nodes. Its interpolation function takes the form

$$y_i(x) = a_i x^3 + b_i x^2 + c_i x + d_i \quad (3)$$

where $y_i(x)$ is the interpolation function for the segment from (x_i, y_i) to (x_{i+1}, y_{i+1}) . For data with n points, the coordinates of data points provide $2n$ equations, the first and second order continuity conditions provide $2n-2$ equations, and the remaining two equations are given by the boundary slopes.

The curvature of a beam under small deformations can be approximated by the second order derivative of its displacement $z''(x)$ and is proportional to the bending moment $M(x)$, i.e. $z''(x) = M(x)/EI$ where EI is the bending stiffness. In the CRS, $M(x)$ is continuous when there is no concentrated bending moment. Based on the

above analysis, the use of third-order Hermite interpolation to calculate the compression of the boundary servos is a reasonable approximation, because this method guarantees $z \in C^2[x_s, x_e]$.

Next, we discuss the problems faced when using the interpolation control method and their solutions. Starting with a 1D case, let the deformation equation of the CRS be **Supplementary Equation Error! Reference source not found.**, i.e., $z(x) = \varphi(x, x_i) = h[1 + \cos 2\pi(x - x_i)/l]/2$, in which $x_i - l/2 \leq x \leq x_i + l/2$, h is the wave amplitude, l is the wave length, and x_i is the peak position. Let the pixel spacing be d . When the displayed wavelength is close to the pixel point spacing, the interpolation function does not fit the target waveform well due to the low sampling rate, shown in Supplementary Fig. 21a. As a result, a biased estimate of ΔL is obtained, as shown in Supplementary Fig. 21b when the wave peak is moving between pixels. The effect of the normalized display wavelength l/d and normalized amplitude h/d on the prediction error of ΔL is analyzed. Define the estimation error rate of ΔL as

$$e_{\Delta L} = \frac{\int_0^d |\Delta L_t - \Delta L_e| dx_i}{\int_0^d \Delta L_t dx_i} \times 100\% \quad (4)$$

where ΔL_t is the target compression and ΔL_e is the estimated compression calculated from the interpolation curve. This definition can be considered as the ratio of the area of the red portion to the portion below the blue straight line in Supplementary Fig. 21b. The error surface of the interpolation algorithm for different normalized wavelengths and amplitudes can be obtained by numerical calculations, as shown in Supplementary Fig. 21c. It can be seen that the error becomes evident for low amplitudes and short wavelengths.

The minimum normalized wavelength that the CRS can display in this paper is $l/d = 3$. Accurate estimation of ΔL at this wavelength is important. According to the

previous analysis, the interpolation algorithm produces an estimation error of about 10% when $h/d = 1$, as shown in Supplementary Fig. 22a. When higher accuracy is needed, it is necessary to increase the rendering resolution in the controller memory. For instance, in Supplementary Fig. 22b, the number of pixels in this array is doubled, and an auxiliary rendering node is added between each pair of pixel points to more accurately calculate the amount of compression. In Supplementary Fig. 22c, the corresponding error is shown to have reduced to one tenth. In two dimensions, this algorithm requires four times the amount of computation, which is often affordable considering the small array size of existing haptic displays compared to computer screens.

For wearable devices with the CRS, the total length of the beam on a spatial curve is often required. For instance, the blue spatial curve shown in Supplementary Fig. 23a, with its projection on xOy and yOz represented by black lines, consists of four independently controlled pixel points. The arc length s can be used as a parameter to represent an arbitrary spatial curve, with the arc length corresponding to the data points defined as s_i for each data point. Subsequently, $x_i \sim s_i$, $y_i \sim s_i$, $z_i \sim s_i$ are independently interpolated as shown in Supplementary Fig. 23b. With the arc length parameter s , the directional vector of the curve at each point is given by $(dx/ds, dy/ds, dz/ds)^T$. Interpolation curves can be uniquely determined by considering the tangent direction conditions at the start and end points as shown in Supplementary Fig. 23c by a red curve. To use this method, it is necessary to calculate the arc length s_i of the pixel point on each skeleton when it is not deformed. s_i for each pixel do not change after deformation by noting that the length change of the skeleton is negligible compared to its whole length. If increased control accuracy of the boundary servos is desired, a double rendering resolution is recommended.

Supplementary Fig. 21 Interpolation control method of CRS. **a** Interpolation of a pixelated wave. **b** Relation between the error of the estimation of ΔL and the peak position x_i . **c** Contour plot of $e_{\Delta L}$ in the l/d and h/d space.

Supplementary Fig. 22 Double-resolution rendering of CRS. **a** $e_{\Delta L} \sim h/d$ curve when $l/d = 3$. **b** Double-resolution rendering. **c** $e_{\Delta L} \sim h/d$ curve when $l/d = 3$ after applied double-resolution rendering.

Supplementary Fig. 23 Interpolation in spatial data point. a Spatial curve and data point selection. **b** Separate interpolation of x_i , y_i , and z_i . **c** Interpolation result for the spatial curve.

Comment 1.5: This would also provide meaningful information on the limitations of the proposed work in terms of supported volume, size, length, etc.

Authors' reply: Agreed. We have added a discussion on the limitations in the main text and Supplementary Information. The mechanical limitations of the CRS primarily stem from the yield stress of the skeleton's material and the travel distance of the boundary

servos. The maximum curvature of the skeleton is limited by yield stress and geometric design limit, while the total length of the displayed shape is constrained by the travel distance of the boundary servos.

However, the size of the CRS does not limit its applications. From a mechanical perspective, the no-collapse condition of the CRS in contact with the skin is scale-free. From a manufacturing perspective, the beam is a component with a simple shape and can be easily miniaturized. Consequently, the design of the CRS can be utilized in haptic devices of various sizes.

(Page 9 of the main text)

...A more detailed discussion of the mechanical behavior of the CRS is provided in Supplementary Discussion 5. Besides, it should be noted that the CRS is subject to additional constraints due to the material properties and the travel distance of the boundary servos (see Supplementary Discussion 8).

(Page 15 of Supplementary Information)

Supplementary Discussion 8 Mechanical limitation of Continuity Reinforcement Skeleton

Although CRS can enhance the display continuity of a haptic device, it also introduces additional motion constraints. These mechanical limitations arise mainly from the travel of the boundary servos and the limitations of the beam material, which are discussed separately below.

In the previous section, a method for controlling the compression of CRS boundary servos was provided. When both ends of the servos are set, the range of variation for the CRS length cannot be greater than twice the servo travel u_{\max} , that is

$$\max(\Delta L_e) - \min(\Delta L_e) \leq 2u_{\max} \quad (5)$$

When designing the CRS system, it is important to determine the maximum and minimum lengths of the CRS with a known pixel spacing d , maximum pixel travel h , and the shape of the base. Due to the complexity of interpolating or solving the real CRS motion model, it is difficult to determine the pattern corresponding to $\max(\Delta L_e)$ and $\min(\Delta L_e)$. In the following, we present an approximate estimate to facilitate the design.

Supplementary Fig. 24 Travel Limit Schematic. **a** Shape of minimum length in straight base. **b** Shape of maximum length in straight base. **c** Shape of minimum length in circle base. **d** Shape of maximum length in circle base. **e** Minimum CRS length estimation in wearable device. **f** Maximum CRS length estimation in wearable device.

Consider the extreme case of $h \gg d$. For planar bases, the shortest and longest situations are shown in Supplementary Fig. 24a and b, respectively. A CRS with $2n+1$ pixel points requires a boundary servo travel distance of about $2nh$. And for a circular base with $2n$ pixels whose radius is r , the shortest and longest situations are shown in Supplementary Fig. 24c and d. The required travel of boundary servo here is about $2nh+2\pi h$.

Based on the above analysis, we can assess the feasibility of equipping CRS on the human body. The radius of the lower arm of an adult male is about 40mm. Around the arm, we set 39 pixels, with a pitch of about 6mm. Assuming a maximum height of 3 mm for each pixel, we can use the interpolation method to calculate the required action distance. Compared to the length of CRS with all pixel heights at 0, the shortest CRS requires the boundary servo to stretch by 19 mm, while the longest CRS requires the boundary servo to compress by 35mm (Supplementary Fig. 24e and f). Considering that there are two servos for each skeleton, the total travel distance required for a single servo is 27mm, which is an acceptable value.

The yield strength of the beam material restricts the maximum bending curvature of the skeleton. According to the formula for the bending normal stress in the cross-section of a beam, when the yield stress of the material is σ_u , its curvature κ at any point cannot exceed

Comment 1.6: There are some places in the paper where technical implementation details are missing. For example, the detail of the mechanical structure should be demonstrated for clarity in Figure 2.

Authors' reply: In the revised paper, we have added detailed information about the direction of the servos to enhance visualizing the function of the mechanical structure.

(Fig. 2a has been updated)

Fig. 2a An 1D continuity reinforcement haptic device.

Comment 1.7: Although this paper is in nature communication, as an HCI researcher, I find that Figure 5 with VR context misses lots of information. For instance, I would like to find out what method has been applied to reduce the haptic effect with the CRS device. The system diagram would be necessary to explain how the overall system works as a VR haptic system.

Authors' reply: We would like to thank the reviewer for pointing out the missing information. We have modified Fig. 5c to focus on explaining how the VR system functions, instead of delving into the technical details.

(Fig. 5c has been updated)

Fig. 5c Haptic-visual combined system diagram.

Comment 1.8: Moreover, the algorithm block on how the CRS haptic display was rendered would be needed in the ‘Methods’ section rather than simply explaining how Meta Quest and MATLAB were configured.

Authors’ reply: Agreed. We have provided additional explanation of our haptic rendering algorithm in the Methods section.

(Page 16 of main text)

...When the fingertip approaches the cyan area that represents recognition in **Error! Reference source not found.**, the VR program sends the position of the finger relative to the center of the CRS recognition area (x_f, y_f, z_f) to MATLAB via UDP communication for conversion into haptic signals, where the z_f direction is perpendicular to the recognized region. The shape of the display of the 2D CRS is determined by the equation

$$z = \frac{1}{2} z_f \left[1 + \cos \left(2\pi \sqrt{(x-x_f)^2 + (y-y_f)^2} / 90 \right) \right] \quad (6)$$

, where $\sqrt{(x-x_f)^2 + (y-y_f)^2} \leq 45$. The haptic rendering thus takes the shape of a rotating sinusoidal surface with a diameter of 90 mm whose height z_f is determined by the depth of the press and position (x_f, y_f) is determined by the fingertip’s position. The compression control algorithm for the boundary servos is calculated directly from the equation of the surface, referring to the Supplementary Discussion 7. By continuously stroking in the recognition area, the movement of the finger can be displayed continuously on the 2D CRS...

Comment 1.9: There have been various previous works on shape-changing display with flexible covers shown below. I would recommend the authors to include

these works and highlight the contribution of the proposed work.

Authors' reply: Agreed. We would like to thank the reviewer for recommending the references and have added them to the revised paper. The corresponding articles are marked in red in the main text and reference list.

(Page 3 of main text)

...Notable studies addressing this issue include forming surfaces with **origami**^{34,49}, **kirigami**⁵⁰, or **semi-solid surfaces**⁵¹, offsetting the unit³¹, developing surfaces with negative Poisson's ratios⁵², and manipulating magnetic fluids⁵³....

Reviewer #2

General comment: This work presents a continuity reinforcement skeleton (CRS) that can be placed on top of a pin array used to display shape. This skeleton allows for the smoothing of motions, and allows the haptic display to obtain smooth shapes and continual motion of a peak point along a beam. It overcomes the fundamental challenge of pin arrays, that for better approximations of smooth shapes more and more pins are needed. CRS is applied to a row of actuator and a single beam, an array of hexagonal pins, and a curved figure. The hardware presented is a novel contribution, but some additional analysis would further strengthen the contributions of this paper.

Authors' reply: We are grateful to the reviewer for reviewing our paper.

Comment 2.1: The hardware design consists of a set of thin beams that go over the top of the pins, constrained at each pin with a loop of fishing line. Additional servos at each end of the beam control the overall length of the beam. This hardware design is a compelling solution. It is able to render smooth shapes and continuous motion. However, whereas conventional pin arrays cannot create smooth shapes, the pin array with the CRS cannot produce shape with sharp edges, which may be desirable in haptic displays. Adding this CRS enables new abilities, but at the expense of others. There may be cases where rendering the edge of a surface, rendering braille characters, or rendering a point contact are desirable for haptics.

Authors' reply: Pixel haptics is usually preferred in Braille displays, mainly because continuous motion is not necessary for this type of task. If continuity is important, CRS is more suitable as it has lower distortion at the same pixel density.

Braille displays only need to present information at fixed pixel points to create specific patterns, without requiring dynamic patterns. However, the purpose of VR

haptics is to simulate the sense of touch when interacting with virtual objects. Continuity in haptics thus becomes crucial during actions like stroking or exploring the shape of an object. Unlike sharpness, which relies on pixel edges that cannot move with the rendered graphic (as depicted in Fig. R1a), contact peaks often do not align precisely with individual pixel points. Consequently, even if the edge is sharp, it is difficult to simulate the feeling of a blade scraping against the skin.

When continuity is important in a haptic system, CRS consistently increase the details of the display. As shown in Fig. R1b, the distortion of CRS is lower than that of pixel-only displays with the same d/l , which means that CRS has a higher degree of reproduction of the displayed graphics.

In addition, the misconception that CRS cannot render sharp edges may stem from the size and range of motion of the actuators we use. The mechanical model and no-collapse conditions given in the paper for the CRS are scale-independent. As shown in the distortion curves in Fig. R1b, the CRS can display more complex and sharp details if combined with a higher density actuator (let d/l be as small as possible).

Fig. R1. Resolution of CRS. **a** The movement of a sharp edge move in pixel-based haptic. **b** Distortion curves of CRS and pixel-based haptic.

We have also included extra explanations in the Discussion section of the main text.

(Page 13 of main text)

...various existing pixel-based haptic displays by incorporating beams on the contact interface. The simple structure of CRS allows for fabrication in various sizes and easy miniaturization. Moreover, the no-collapse condition remains intact when the skeleton is proportionally scaled-down. When combined with a haptic device with higher pixel density, CRS can depict more detailed and sharper shapes. CRS also expands...

(Page 14 of main text)

...The display effect of the sharpest edges in complex graphics can be estimated using the distortion theory (See Supplementary Discussion 4). Finally, this evaluation criterion reveals the average display quality of the pattern when it spans the entire domain, as pressure peaks may not always align precisely with the pixel point and must be able to move. For a haptic device capable of displaying sharp edges, high continuity means that it can simulate the feeling of a blade scraping against the skin, while low continuity means that those sharp edges cannot move.

Comment 2.2: I would also be interested to know more of the limits of the device. How does the CRS affect the maximum height that can be achieved by the peak? There are cases as in figure 3 where the peak of the beam extends beyond the height of the pins. What is the additional force required from the pins to deform the beam? Is there a case where the strain limit in the beam could become the limiting factor?

Authors' reply: The skeleton does not overburden the servos at pixel points because the compressions at the ends of the skeleton cause it to buckle. Meanwhile, the pixels restrict the shape of the buckling skeleton. Our analysis shows that when the display shape is fixed, the pressure on the pixel points decreases as the compression of the boundary servos increases. The detailed discussion is provided below.

Furthermore, we have included a discussion on the limitation of the yield stress of CRS's material in Supplementary Discussion 8. Please also see our reply to Comment 1.5.

(Page 9 of Supplementary Information)

It is also desirable to solve the support reaction of the boundary servos and pixel servos. Taking the first two terms of the serial solution in Supplementary Equation **Error! Reference source not found.**, we get

$$y = 4Pl^3 \left[k_1 \left(1 - \cos \frac{2\pi x}{l} \right) + k_2 \left(1 - \cos \frac{4\pi x}{l} \right) \right] \quad (7)$$

in which

$$k_2 = \frac{3}{512\pi^4 EI - 288\pi^2 d^2 N + 486\beta d^4} \quad (8)$$

However, because $N = N_{cr}^{(1)} = (16\pi^4 EI + 3\beta l^4) / 4\pi^2 l^2$ is obtained by letting $n^4 16\pi^4 EI - 4n^2 \pi^2 l^2 N + 3\beta l^4 = 0$, the value of k_1 should be determined by the geometric constraints. The compression of the boundary servos should be $\Delta L = \Delta l + \Delta L_0$, in which Δl is the reduction in the total length of the beam due to the axial force, and ΔL_0 is the length increment due to the off-plane deformation. Note that when $\Delta \gg 1$, $\Delta l = N_{cr}^{(1)} L / EA \approx \pi^2 b^2 L / 3l^2$ is material-independent, with L and b being the length and thickness of the beam, respectively. The change in beam length Δl is negligible for CRS with high Young's modulus and small thickness. Considering

the constraints of the support points and servos compression on both sides, we get

$$\begin{cases} \Delta L \approx \frac{1}{2} \int (y')^2 dx = 16\pi^2 P^2 l^5 (k_1^2 + 4k_2^2) \\ \frac{3}{4} h = 4Pl^3 \left[k_1 \left(1 - \cos \frac{2\pi}{3} \right) + k_2 \left(1 - \cos \frac{4\pi}{3} \right) \right] \end{cases} \quad (9)$$

The equation is simplified to obtain

$$\begin{cases} \Delta L = 16\pi^2 P^2 l^5 (k_1^2 + 4k_2^2) \\ h = 8Pl^3 (k_1 + k_2) \end{cases} \quad (10)$$

The following results on the pressure on the pixel point and the shape of the deformed beam can be obtained by combining Supplementary Equations (8), (9), and (10).

$$\begin{cases} N = N_{cr}^{(1)} = \frac{16\pi^4 EI + 243\beta d^4}{36\pi^2 d^2} \\ k_2 = \frac{3}{512\pi^4 EI - 288\pi^2 d^2 N + 486\beta d^4} \\ k_1 = \frac{12d\Delta L + 2\pi h \sqrt{15d\Delta L - \pi^2 h^2}}{\pi^2 h^2 - 12d\Delta L} k_2 \\ P = \frac{h}{216d^3 (k_1 + k_2)} \end{cases} \quad (11)$$

Supplementary Fig. shows the relationship between the reaction force and the compression of boundary servos, with $E = 210\text{GPa}$, $\beta = 210\text{kPa}$, $d = 10\text{mm}$, and $h = 10\text{mm}$. When using the direct control method to determine ΔL (see Supplementary Discussion 7), the reaction force on the pixels is shown as the circle in orange. If the force on the pixels need to further reduce, the control algorithms can be fine-tuned to increase the compression. If the actuator of the haptic device is force-controlled, we can use Supplementary Equation (11) to calculate the amount of required force to achieve a specific target shape.

Supplementary Fig. 17 Relationship between pixel reaction force and boundary servo compression.

Comment 2.3: It is interesting that there is no discussion of human perception in this paper. The authors have instead discussed that they instead use mathematical metrics, but not ones that consider human perception. This is logical considering the focus of the device is on the display, but the contribution would be strengthened if these mechanical marks could be shown to correlate with perceptual metrics. It seems to me that the ability to display sharp edges still be very valuable for some perception tasks.

Authors' reply: Agreed. We have included human computer interaction (HCI) experiments. CRS-based handwritten digit identification experiments have been added to the Supplementary Discussion 9. Compared to pixel-based haptic, CRS can increase the accuracy of identification from 50.3% to 67.8%. We have detailed the HCI experiment in our reply to Comment 1.1. We have also discussed the issue of sharp edges in our reply to Comment 2.1.

Comment 2.4: In figure 3 on the phase diagram, the amount of experimental data does not make the transition clear. Having more data to analyze this phase transition would be a valuable addition

Authors' reply: We have performed additional tests and increased the number of experimental data points in the phase diagram. We selected three different silicones, three skeleton materials with different Young's moduli (aluminum, copper, and steel), three different pixel pitches, and three different thicknesses, resulting a total of 81 sets of experiments compared to 18 sets in the original paper. All experimental results are within the defined domain of the phase diagram and are labeled in the figure. The good agreement between the experiment and theory provides added confidence in the theory.

(Fig. 3d has been updated)

Fig. 3d Phase diagram of collapsing, where the triangle and circle markers represent experiment result.

Comment 2.5: The computational analysis focuses on determining when the beam will buckle. Is a computational model also used to control the position of the boundary servos and heights of the pins. It is unclear to me how the position of the boundary servos are controlled.

Authors' reply: We have provided the detailed control algorithm in the revised paper. Please see our reply to Comment 1.4.

Comment 2.6: The evaluation primarily uses digital image correlation, which capture well the performance of the device. The authors, in both modeling and experiments, consider the beam applying force to the a silicone structure to represent the human body, which is a good approximation.

Authors' reply: We would like to thank the reviewer for appreciating our effort.

Comment 2.7: The videos and figures are well done and very explanatory. If possible, adding a scale reference in the videos would be useful.

Authors' reply: Agreed. We have included scale references in the updated videos and figures.

Reviewer #3

General comment: This paper proposes and evaluates a method to create 1D and 2D haptic displays augmenting traditional haptic-pixel displays using a Continuity Reinforcement Skeleton (CRS). In a nutshell, the CRS is defined by a series of steel beams that connect haptic-pixels enabling a continuous shape rendering. The beams are also actuated by end effectors to improve the response of the system.

The paper also discusses future works that could extend the CRS approach, by providing additional haptic cues, such as tangential forces or vibrations. These highlights additional benefits over traditional haptic-pixel displays.

Haptic displays is an active research field which deal with the challenge of proposing effective and simple systems to deliver rich kinesthetic and tactile feedback. The proposed system provides a modular and cost effective approach to deliver cutaneous feedback during the exploration of virtual surfaces.

Authors' reply: We sincerely appreciate the reviewer for taking the time to review our article.

Comment 3.1: However, authors mainly use the system to display simple shapes, it remains unclear how the system could render more complex shapes in which multiple pixels are actuated arbitrarily.

Authors' reply: The developed CRS is capable of displaying complex shapes. We added new cases to the second half of the revised Video S2, in which we reduced the pixel spacing to half of the original one and show the display of non-sinusoidal waves. The experiments show that even non-sinusoidal waveforms can also move continuously with the help of the CRS, as shown in Fig. R2. In fact, the haptic device equipped with

CRS can display more complex shapes with the density of the actuator and its travel distance increasing.

Fig. R2. Display of a non-sinusoidal waveform with the CRS.

Comment 3.2: Another potential limitation is that the system is unable to render sharp edges. This could be mitigated by the increase of the density of the pixel display, although reducing the gain of the use of the CRS.

Authors' reply: It is true that some currently available pixelated haptic devices do display sharp edges. However, these sharp edges can only appear at fixed positions and cannot move. Such haptic displays are often used in situations where the movement of contacts is not important. For instance, Braille displays only need to show sharp, specific patterns without the need for movement. In contrast, during VR interactions, pressure peaks may not always align precisely with the pixel point and must be able to move. Therefore, it is essential to consider the performance of the waveform while moving across the entire domain. Consequently, adding CRS to the haptic system can lead to lower distortion compared to a pixel-only display at the same density.

On the other hand, higher pixel density does not reduce the usefulness of CRS, but instead enhances the ability to display sharp edges. If the pixel pitch d is reduced, the

minimum wavelength l that can be displayed is also reduced under the same distortion requirement. As demonstrated in Fig. R3, when a triangular wave needs to be displayed, increasing the pixel density brings the shape closer to the desired triangular wave and sharpens its peak.

We have also clarified this question in the Discussion section of the main text.

Please also see our reply to Comment 2.1.

Fig. R3 Display of a triangular waveform with the CRS of different pixel densities.

Comment 3.3: Although the proposed VR scenario is compelling, the added value of the VR seems limited and it remains unclear how the pixel-based display could be easily worn by users.

Authors' reply: The main purpose of the VR scenarios here is to demonstrate VR interaction when haptic devices do not need to track users' movements. For instance, a user can place his/her arm on a surface to feel a distant touch.

The current limitation preventing it from being wearable is the size of the actuators. Combining CRS with a thin, lightweight, and pixelated haptic device (e.g., the hydraulically increased haptic device of Leroy et al.⁶) would make the CRS device wearable. The control algorithm for the CRS has been provided in our response to

Comment 1.4, and it is hoped that these analyses can assist researchers in integrating CRS with other haptic devices.

Comment 3.4: Finally, one aspect that is not discussed in detail on the paper, is how the gaps in the 2D version of the device are covered, and whether this is actually required. This doubt is raised due to the lack of explanation of the intended use of the device, is the user interacting with it, or it is always in contact with the surface of the skin. From the content of the paper, it seems the later, but it should be clarified.

Authors' reply: The gaps in 2D CRS can be minimized but cannot be completely covered. The gaps become progressively smaller with increasing pixel density. Their final appearance is similar to a kitchen sieve. If we were to fully enclose the entire surface with a thin sheet, the sheet would experience significant stress when displaying an undevelopable surface. In contrast, the frame design of CRS enhances deformability of the surface and reduces stress on the actuator.

Additionally, CRS can be used to realize the interpolation of haptic information between pixel points, regardless of whether it is wearable or not. When combined with thinner actuators, CRS devices can be worn and maintain contact with the surface of the skin.

(Page 10 of main text)

...Fig. 4b illustrates how the CRS device displays a non-developable surface that cannot be obtained from a plane without tearing, creasing, and stretching. **Due to the frame structure of CRS, the beams only undergo bending deformation, which prevents the generation of large internal stress and enhances deformation capacity.**

Comment 3.5: The paper discusses the requirements of the CRS in terms of shape continuity and the mechanical model. The proposed data shows the viability of the system and the forces capable of delivering.

Authors mainly discuss their results against discreet haptic-pixel devices, and in particular, all evaluations are conducted considering the same prototypes, removing the CRS structure. These evaluations, although they ensure that both systems are equivalent in terms of pixel density, are strongly biased due the large spacing between haptic-pixels. Traditional haptic-pixel displays will exhibit higher density to reduce the gap between pixels. These generates unrealistic configurations for haptic-pixel displays. This somehow generates a biased comparison in which the CRS display is always superior. Comparing the system with higher density displays would have provided a fairer description and authors could have discussed the density increase to achieve comparable results.

Authors' reply: Thank you for your comment and we need to clarify three points on this issue:

1. CRS is an interpolation interface designed for pixel-haptic displays, not a particular kind of haptic device. Therefore, its effectiveness should be evaluated by comparing CRS with a pixelated haptic device of the same pixel density for the purpose of controlling variables. Traditional pixel displays may be able to exhibit higher densities, but it can also be equipped with CRS to present a better display effect.

2. CRS can be miniaturized. In particular, the value of the no-collapse discriminant does not change when the size of the skeleton is proportionally scaled down, which means that it can be implemented for higher pixel densities. In our reply to Comment 3.1, the pixel pitch is reduced by half to 15mm, and the CRS can still work well. The

lower pixel density in this study is due to the size limitation of the available actuators and not the limitation of the CRS design itself.

3. Regarding the large pixel spacing employed in this study, square pixels and hexagonal pixels with no spacing have been used to calculate the distortion of 2D pixels, shown in **Error! Reference source not found.**. Notably, the shape distortion curve of the no spacing haptic remains higher than that of the CRS. If we fix the shape distortion, the CRS is equivalent to quadrupling the face density, according to the curve of the relationship between shape distortion D_s and d/l (where d is pixel pitch, l is the displayed wave length).

Supplementary Fig. 4 Fitting curve for the shape distortion of different pixel shape.

Comment 3.6: Two aspects not discussed in the paper raised my attention. First, authors do not discuss the coupling between the “haptic-pixels” and the beams. Multiple activations of pixels attached to the same beam might generate too much tension.

Authors’ reply: We have added a discussion on the coupling between the “haptic-pixels” and the beams, in particular, the CRS reaction forces imposed on pixels in the revised

Supplementary Discussion 5. Actually, the CRS uses boundary servos to compress the skeleton and induce buckling. The shape of the buckling is controlled by the pixels to achieve interpolation. By increasing the compression of the boundary servos, the force imposed on pixels can be significantly reduced. Please also see our reply to Comment 2.2.

Comment 3.7: Second, authors do not discuss the latency of their system. Considering that is intended for real-time use cases, latency should be assessed.

Authors' reply: Latency is essential in VR. We have included the latency of our VR system in the revised Methods and Supplementary Information.

(Page 17 of main text)

...and the height of the bump can be controlled by the depth of the press in the recognition area. The latency between the VR system and the CRS is about 440ms (See Supplementary Discussion 11 for measuring method).

(Page 19 of Supplementary Information)

Latency is essential for haptic device. Here, we set the camera to high-speed mode and position the user and the haptic device within the same frame, as shown in **Error! Reference source not found.** Initially, the user's hand rested at the center of the virtual character's hand, and then the user was instructed to start pressing. The delay time is measured by the time difference between the start of the hand movement and the response of the haptic device. The latency here is about 440 ms.

Supplementary Fig. 31 Latency measurement of the VR haptic system.

Comment 3.8: The analysis is sound and carefully assesses the continuity of the proposed prototypes evaluating the forces and breaking points of the device. Precisely, two major assessments are provided, the deformation of the connecting beams, and the pressure that the beams can support from external forces (e.g. skin).

Authors provide enough information to replicate the proposed system and conduct the evaluations.

Authors' reply: We are grateful to the reviewer for his/her appreciation of our work.

Comment 3.9: The title could be a bit more informative, such as “Continuity Reinforcement Skeleton for Pixel-Based Haptic Displays”

Authors' reply: Thank you for the advice. The title has been changed, as suggested.

Comment 3.10: (ln 20) Authors talk about "a novel perspective compared to traditional psychological assessment", I do not see the novelty on their assessment.

(ln 22) The system has not been tested in VR with real users, so the statement is weak and evidence is missing.

(ln 37) This statement is false, a wide range of haptic devices exist to generate pressure and forces (already discussed in the previous paragraph of the paper). Maybe authors wanted to state that "Most haptic devices that rely on matrix of pixels to display pressure,..."

(ln 85) In several parts of the paper, authors talk about the "user's skin" it remains unclear the type of interactions that are considering.

(ln 88) "it minimizes interference with haptic feedback while ensuring robust connectivity" this sentence is not justified.

(ln 93) In several sentences authors talk about "haptic experience" or "user experience", it remains unclear what authors are referring with it. Considering the evaluation conducted authors, it should refer to the quality of the generated shape.

Although the paper is easy to read, it will benefit from a full English proof-read.

Authors' reply: We have revised the inappropriate statements in the text as follows.

(ln 20-22) Efforts are made to quantify haptic display quality using geometric, mechanical, and psychological criteria. The development and integration of one-dimensional (1D), two-dimensional (2D), and curved CRS devices with virtual reality systems highlight the impact of CRS on haptic display, showcasing its potential for improving haptic experience.

(ln 37) Most haptic devices rely on a matrix of pixels to display pressure, with each pixel moving independently to create tactile sensations.

(In 85) To address the loss of haptic information typically experienced between pixels, the 1D CRS device (Fig. 2a) incorporates a 0.1-mm-thick steel beam on the pixels.

(In 88) The choice of fishing line as a connector serves a dual purpose: it reduces interference with the haptic interface while ensuring an adequate connection strength.

(In 93) These boundary servos control the total lengths of the beams in the display area to satisfy the geometric constraint.

Comment 3.11: References that can be of interest of authors:

Sturdee, Miriam, and Jason Alexander. "Analysis and classification of shape-changing interfaces for design and application-based research." *ACM Computing Surveys (CSUR)* 51.1 (2018): 1-32.

Zhu, L., Jiang, X., Shen, J., Zhang, H., Mo, Y., & Song, A. (2022). TapeTouch: A handheld shape-changing device for haptic display of soft objects. *IEEE Transactions on Visualization and Computer Graphics*, 28(11), 3928-3938.

Wang, Jiabin, et al. "A novel shape-changing haptic table-top display." 2017 International Conference on Optical Instruments and Technology: Optoelectronic Imaging/Spectroscopy and Signal Processing Technology. Vol. 10620. SPIE, 2018.

Boem, Alberto, and Giovanni Maria Troiano. "Non-rigid HCI: A review of deformable interfaces and input." *Proceedings of the 2019 on Designing Interactive Systems Conference*. 2019.

Authors' reply: The suggested references, offering different insights into haptic devices and the key efforts in this area, have been included in the revised paper. The corresponding articles are marked in red in the main text and reference list.

(Page 5 of main text)

...with various devices based on different principles, including deformation-driven^{18,28,31-35}, ultrasonically-driven^{36,37}, and electrotactile^{38,39}.

Most haptic devices rely on a matrix of pixels to display pressure, with each pixel moving independently to create tactile sensations^{19,22,40-43} ...

References

1. Biswas, S. & Visell, Y. Emerging material technologies for haptics. *Adv. Mater. Technol.* **4**, 1900042 (2019).

Responses to reviewers' comments

In the following, the authors' replies, summarized in a one-to-one manner, are shown in **blue**, with the changes made in the revised manuscript denoted in **red**. In addition, while using the original numbering of figures/references in the manuscript (e.g., Fig. 1, ref. 1), we add the prefix "R" for those presented in this response letter (e.g., Fig. R1, ref. R1).

Reviewer #1

General comment: Hi. I have read the authors' revisions. I found that there is a message "Error! Reference source not found" throughout the revision letter. (I am not sure what this relates to) Overall, I liked the thorough revision reflected by the authors. Still, I have below comments. Below are my thoughts on the revised context.

Authors' reply: We are grateful to the reviewer for his/her invaluable time spent reviewing our paper. We are sorry for the "source not found" problem that occurred after uploading our manuscript to the system. We have corrected the error.

Comment 1.1: I liked the additional user study with the proposed system. However, I also felt that the current reported performance is low (<70%). Could we confidently say use this system as claimed with this user performance? I know it is better than the pixel-based approach baseline; the authors should provide a rationale for this relatively low performance and how it can be overcome in future iterations. This should be explicitly mentioned in the Limitation section to prevent any overclaim.

Authors' reply: The reason for low performance is that the handwritten word is small compared to the pixel size. This is deliberately designed to show the advantages of continuity reinforcement skeleton (CRS). As mentioned in Supplementary Discussion 9,

the digits have an average width of 14.8 mm and an average height of 24.3 mm, while the pixel pitch is 30 mm. Thus, users need to capture sub-pixel level contact movement traces at lower pixel densities, resulting in low recognition rates. On the other hand, some numbers with similar strokes, such as “3” and “5” are not easily distinguishable from each other, which makes it even harder to identify the digits. We agree to add extra explanation in the Limitation section. The following changes have been made to the revised paper.

(Page 12 of main text)

.....As shown in the experiment given in Supplementary Discussion 9, the CRS device, when compared to a pixel-haptic, can increase the accuracy of digit identification from 50.3% to 67.8%. The lower recognition rate here is mainly due to the smaller size of the digits compared to the pixel density. The average height and width of digits are only 0.81- and 0.49-pixel pitch.

(Page 14 of main text)

In this paper, the discussion on CRS systems focuses on the display of pressure. In order to display complete haptic information, further work can focus on displaying higher pixel densities, tangential force, vibration, and temperature. Although CRS can improve the digit identification rate, pixel density still needs to be increased to display more detailed tactile information to further improve performance. Then, the beams in the CRS can move along their length direction...

Comment 1.2: For supplementary Discussion 7, I am glad that the authors provide a generalized method to expand the form factor of the proposed system. However, I

would like the authors to clarify when to use the direct control and interpolation methods with integrated figures if possible.

Authors' reply: It should be clarified that the direct control method can only be used to calculate the amount of compression if the equation of the corresponding curve of the CRS is known, and is therefore only applicable in the testing phase. In practice, we only have information about the height at the pixel points, not the equation of the corresponding curve of the CRS. Therefore, we can only estimate the amount of compression through interpolation. We have emphasized this issue in our revised version and added an additional Supplementary Table to explain the features of the three methods.

(Page 12 of Supplementary Information)

Note that **unlike** the previous formulas in integral form, the above calculation of lengths does not require derivation. This formula is also applicable to CRS on curved surfaces if the discretization is done in a reasonable way. **It should be noted that the direct control method is only suitable if the equation of the CRS is known, which is often not available in practical VR environments.**

(Page 15 of Supplementary Information)

... a double rendering resolution is recommended. **To facilitate the understanding of the three compression calculation methods in this paper, we listed their advantages and disadvantages in Supplementary Table 2.**

Supplementary Table 2 Comparison of the three control methods

Control method Item	Direct control method	Interpolation control method without auxiliary points	Interpolation control method with auxiliary points
Advantages	Fast, accurate	Able to calculate compression if the heights of pixels are known	Accurate, able to calculate compression if the heights of pixels are known
Disadvantages	Require the equation of the CRS curve	Large error when displaying details of smaller wavelengths	Need additional height information for auxiliary nodes

Comment 1.3: For Supplementary Fig.22, what is the role of “Auxiliary node”? I would also like the author to integrate Fig. 21 and Fig. 22. It would be even better if the authors could include the explanation of the direction control algorithm in this figure as well as mentioned earlier.

Authors’ reply: In order to reduce errors when calculating compressions using only the displacement information at pixel points, the displacement information of the auxiliary nodes can be utilized to achieve higher density interpolation when rendering haptic information. In short, the displacements at the auxiliary nodes do not correspond to

pixel points, but are used solely for compression calculation. We have merged the mentioned two figures into a new one. Here is the revised version.

(Page 31 of Supplementary Information)

Supplementary Fig. 1 Interpolation control method of CRS. **a** Process to calculate the compression of CRS. **b** Relationship between the error of the estimation of ΔL and the peak position x_i . **c** Contour plot of $e_{\Delta L}$ in the l/d and h/d space. **d** Comparison of $e_{\Delta L} \sim h/d$ when $l/d = 3$ without and with an auxiliary node.

Comment 1.4: The limitation regarding volume, size, and length should be explicitly mentioned in the body text rather than only explained in the supplementary.

Authors' reply: Agreed. We have included a separate paragraph in the revised paper to discuss the limitations.

(Page 9 of main text)

Besides, it should be noted that the CRS is subject to additional constraints compared to pixel haptic displays. First, the maximum curvature of the CRS is constrained by the yield strength of the material, which limits the range of pixel movement. Second, since the CRS requires boundary servos compression, its curve length increase needs to be less than the boundary servos' travel distance. For a quantitative discussion of the limitations of the CRS on the range of pixel movement, please see Supplementary Discussion 8.

Comment 1.5: I am worried about the system's current delay, which is well over the typical haptic device requirement (<50~100ms). I think this would be a major limitation in terms of haptic devices, and this should be clearly stated in the body text of the paper. In addition, the authors should suggest how this delay could be reduced in future iterations. In my opinion, the delay is the most critical performance in providing meaningful haptic experience to the user especially when working within AR/VR environment.

Authors' reply: We agree that delay is vital in VR applications. We have adjusted the moving speed of the servos and reduced the delay. Accordingly, the latency of the CRS device itself (when not working with a VR environment) becomes 75 ms. When working with a VR environment, the latency is reduced to 160 ms. The latency can be

further reduced if the VR and control program are integrated and porters with higher speed are used. We have also revised the main text and Supplementary Discussion.

(Page 17 of main text)

...The latency of the CRS device is 75 ms. When the CRS device works with a VR environment, the latency of the whole system is about 160 ms (See Supplementary Discussion 11 for details on the measurement method).

(Page 19 of Supplementary Information)

Supplementary Fig. 30 Latency measurement. a Latency measurement of the CRS device. **b** Latency measurement of the VR-haptic system.

Latency is essential for haptic devices. First, we assessed the latency of the CRS device. The MATLAB control program will display “start” on the screen after sending the control signal to the CRS device, allowing us to measure the latency by calculating the time difference between the appearance of “start” and the response of the CRS device, as shown in Supplementary Fig. 30a. The latency is 75ms. To test the latency

of the VR-haptic system, we set the camera to high-speed mode and positioned the user and the haptic device within the same frame, as shown in Supplementary Fig. 30b. Initially, the user's hand rested at the center of the virtual character's hand, and then the user was instructed to start pressing. The delay time was measured by calculating the time difference between the start of the hand movement and the response of the haptic device. The latency in this case is 160ms.

Reviewer #2

General comment: The authors have made substantial improvements to their report, primarily by adding studies on the users perception of the haptic device, as well as in providing details on the control methodology for the device.

Authors' reply: The reviewer's comments are greatly appreciated as they are very helpful in improving our paper.

Comment 2.1: I am still unclear about the effect of the pixel force. I understand that the majority of the actuation force comes from the servos at the boundary of the device, but isn't there also the case that for some shapes far from the natural buckled modes of the beam, large forces from the pixels are still required. For example, in the case of a steep peak between two constrained pins, it seems like the neighboring pins would still have high force. I did not understand this statement in that regard: "If the force on the pixels need to further reduce, the control algorithms can be fine-tuned to increase the compression."

Authors' reply: Although the mechanical model mentioned in this paper cannot solve the pixel force when displaying an arbitrary pattern, the pixel force can be estimated by a simplified model. Considering the case where the material is subjected to longitudinal forces only, the deformation of the beam can be expressed in terms of a cubic function. Thus, the deformation of the CRS can be approximated by replacing it with a spline interpolation curve based on the continuity condition of the bending moment at the pixel. Then according to the beam equation

$$\begin{cases} \frac{d^2 y}{dx^2} = \frac{M}{EI} \\ M = M_n + F(x - x_i) \end{cases} \quad (R1)$$

a link from pixel height to pixel force can be established. Consider the case in Fig. R1a with a pixel pitch of 15 mm and a peak height of 10 mm. The thickness of the beam is

0.1 mm, the width is 1 mm, and the modulus of elasticity of the material is $E = 210 \times 10^9 \text{ GPa}$. According to Eq. (R1), $F_1 = F_2/2 = F_3 = 0.63 \text{ N}$, which is acceptable for actuators. We have also carried out an experiment to validate the conclusion, shown in Fig. R1b. We made a beam with the same size and boundary conditions as Fig. R1a. Here, one pixel is connected to a force gauge through a fishing line. F_3 in Fig. R1a can be measured by tightening the fishing line so that the displacement of the beam at the corresponding pixel point is zero. The test in Fig. R1c shows $F_3 = 0.64 \text{ N}$ which is consistent with the theoretical solution and is acceptable for our actuators. The precision of the force gauge in Fig. R1c is 0.02 N. Moreover, we have removed the confusing expression in our manuscript.

Fig. R1 Pixel force of the CRS. a Estimation of the pixel force. **b** Model for measuring the CRS pixel force. **c** Pixel force test.

Reviewer #3

General comment: First of all, I would like to thank the authors for their thorough revision and the detailed information provided in the cover letter. Through the revision authors address most of the remarks of the reviewers, and have provided additional details and discussion in the manuscript and the supplementary materials. I believe that the changes introduced are relevant and help to clarify/discuss topics which were briefly addressed or not addressed. In overall, I'm in favor of accepting the contribution, I've just three major points that should be addressed.

Authors' reply: The reviewer's comments are greatly appreciated.

Comment 3.1: While authors have provided additional details on the complexity of the shapes that can be displayed with the CRS device, most of the examples mainly discuss low frequency features (few curves). One interesting feature of pixel displays is their capacity of render high-frequency features, which somehow is limited by the addition of the CRS. Authors should discuss advantages and disadvantages of using the CRS device (e.g., increased continuity at the expense of decreased flexibility on the shapes that can be rendered).

Authors' reply: The CRS device does not limit the complexity of the shape that can be displayed. From the perspective of frequency, the CRS can display features of the same frequency as a pixel display with the same pixel density. For a display with a pixel pitch of d , the smallest wavelength that can be displayed is $2d$. As shown in Fig. R2a, the shape of $2d$ wavelength can also be displayed by a CRS device. Furthermore, when displaying features, the CRS can utilize its own natural interpolation property to fill in the missing tactile information between pixels. We can demonstrate the difference between CRS and pixel displays by showing a human side face in Fig. R2b. Here, the pixel density and height distribution are the same for both devices. The CRS feature is

smoother and easier to recognize. It should be noted that the curve of CRS here is approximated by a cubic spline because the deflection equation is the same for beams subjected to only longitudinal concentrated forces at small deflections.

Admittedly, the CRS does have limitations as well, mainly due to the yield strength of the CRS material and the travel of the boundary compression servos. Please refer to our reply to comment 1.4 and Supplementary Discussion 8.

Fig. R2 Display effect of CRS device. **a** Sine wave with a wavelength of 2 times the pixel pitch, displayed on a CRS. **b** The effect of a pixel display and a CRS device showing the same feature at the same pixel density.

Comment 3.2: The relevance of the VR environment example provided on the paper is still unclear. In its current state, it is merely an illustration. Authors stated in the cover letter "haptic devices do not need to track users' movements", I do not see how this is

specifically linked with the CRS device. To me, it does not bring anything interesting to the paper and could be simply removed. It would have been more interesting to discuss use cases in which the CRS display could be used to overcome the limitations of pixel-based displays.

Authors' reply: We agree that it is important to highlight the features of the CRS rather than using an illustration. We edited our Supplementary Video 5, in which we compare the display effect of writing a digit “9” with pixel and CRS devices to show the continuity reinforcement effect of the skeleton directly. Besides, we also hope to main the VR environment as it allows for the digits identification test and showcases the CRS’s working performance when displaying a moving shape.

Comment 3.3: Authors provided a measurement of the latency of the system, 440ms for haptic interface is really high. It might be interesting to provide details/discuss why such a high latency and how it could be reduced.

Authors' reply: We have adjusted the moving speed of the servo motor. The latency of the CRS device is reduced to 75 ms, and the latency of the whole VR-haptic system is reduced to 160 ms. For more information, please refer to our reply to Comment 1.5. We hope that this information is helpful to you.

Responses to reviewers' comments

The reviewers' comments are greatly appreciated. In the following, the authors' replies, summarized in a one-to-one manner, are shown in **blue**, with the changes made in the revised manuscript denoted in **red**.

Reviewer #3

General comment: I would like again to thank the authors for the revised version. At this stage, I do not have any further comments.

Authors' reply: The reviewer's comments and help are greatly appreciated.

Reviewer #4

General comment: We greatly appreciate the authors' comprehensive response and additional experiments to address all earlier comments and/or concerns. Akin to original reviewers' opinions/comments, this is definitely an interesting and useful design for various VR-HCI systems.

Authors' reply: We are grateful to the reviewer for his/her constructive comments, which are very helpful for improving our paper.

Comment 4.1: In alignment with Reviewer #1's comments, we note that the additional user study on visual-haptic fusion interaction yielded a somewhat low accuracy rate (<70%). The response letter has addressed this. Although we agree with the authors that the limited number size could be somewhat a contributing factor to this slightly lower accuracy rate, we would like to kindly note that there are alternative aspects in which this experiment could be further enhanced.

The size of the numbers in the visual and haptic displays differs significantly. From Supplementary Video 5, the number displayed covers the entire area of the virtual hand, which should be approximately 18 cm in height and 8 cm in width for an average adult hand (source: <https://www.healthline.com/health/average-hand-size>). However, the actual average size of the numbers is only 2.43 cm in height and 1.48 cm in width. Further clarification could be beneficial.

The configuration of the digital identification experiment differs from what is shown in Figure 5, as the average size of the written numbers is significantly smaller. We believe the reason of this modification is worth discussion in the main text.

Authors' reply: As the reviewer has pointed out, the size of the digits has a significant impact on the correctness of recognition. However, the purpose of our digit recognition experiments is to explore the feasibility of the CRS under extreme conditions, e.g., when the size of the digits is close to that of the pixel pitch (about 3 cm). We have added an explanation to the main text regarding why the digit size is smaller than that in VR.

(Page 12, main text)

...which is recorded every 0.5 seconds (See Supplementary Movie 5). As shown in the experiment given in Supplementary Discussion 9, the CRS device, when compared to a pixel-haptic, can increase the accuracy of digit identification from 69.9% to 84.7%. In order to explore the display effect of the CRS under extreme cases, e.g., when the size of the digit is close to that of the pixel pitch, the adopted average height and width of the digit in this test are only about 0.63- and 0.41-pixel pitch, respectively.

Comment 4.2: Stroke order can be another critical factor influencing identification accuracy. From Supplementary Figure 25, we notice that all 10 numbers are written in

a single stroke, which may not be necessarily ideal, particularly for numbers like “4” and “5.” Additionally, the starting points for “8” and “9” could be adjusted. For instance, “8” could start from the center, and “9” could begin slightly lower. We believe these factors can appear very interesting for general readers.

Authors’ reply: The reviewer’s suggestions are greatly appreciated. We have modified the stroke order of the written digits accordingly, re-performed the recognition experiment, and revised the article accordingly. By changing the stroke order, the rates of correctness are greatly improved (i.e., 84.7% for the CRS and 69.9% for the pixel haptic, compared to the corresponding values of 67.8% and 50.3% for the previous designs).

(Page 17 of Supplementary Information)

To investigate the impact of continuity on the haptic display, we measured and compared the accuracy of digit identification using palm haptic with and without CRS. In the experiment, volunteers were instructed to feel the movement of contact (similar to main text Fig. 5b) and identify the corresponding number of the trajectory, see Supplementary Fig. 24. The two haptic devices share the same shape and a pixel pitch of 30 mm. The shapes of the displayed digits from 0 to 9 are shown in Supplementary Fig. 25. For each number, we collected five different handwriting patterns. Here some of the numbers, such as 8 and 9, have different stroke orders, as shown in Supplementary Fig. 25. The numbers 4 and 5 will be written in two strokes, while all other numbers will be written in one stroke. These digits have an average width of 12.3 mm and an average height of 18.9 mm. Notice that the size of the numbers is much smaller than that in the VR demo. This design demonstrates that the CRS can accurately display the peak locations, even in the extreme case when the number size is smaller

than that of the pixel pitch. The shape of the contact point is determined by Supplementary Equation (8), in which $h=10\text{mm}$ and $l=100\text{mm}$, while the peak position is determined by the digits' shape.

A total of eight volunteers (six males and two females) used their left hand for number identification. Each volunteer had 20 learning opportunities, i.e., the numbers were displayed according to the volunteers' request to facilitate their familiarization with the task. The volunteers then performed four sets of identification experiments, with each group recognizing ten random numbers (which could be repeated). Half of the volunteers tested the haptic device with CRS first, then measured pixel haptics, while the other half did so in reverse order.

The confusion matrices shown in Supplementary Fig. 25 were plotted based on the volunteers' identification of numbers. The numbers in the blocks indicate the frequency of the identification results corresponding to specific ground truth. For instance, the "1" at the lower-left corner of the pixel confusion matrix means that the volunteers mistakenly answered "0" while displaying "9" one time. Blue blocks represent correct identification, while orange blocks represent incorrect ones. By improving the continuity of the haptic display, the accuracy in identifying numbers with similar strokes (such as "0" and "6", "7" and "9") increases. The average accuracy of the CRS device is 84.7%, while that of the pixel haptic is 69.9%. By enhancing motion continuity, the accuracy of user identification is improved by about 14.8%.

Supplementary Fig. 25 The trajectories of handwritten digits in the identification experiment.

Supplementary Fig. 26 The confusion matrices of the pixel and CRS haptics in the digit identification.

Comment 4.3: Another minor concern/comment is whether the CRS system would reduce the system’s load-bearing capacity, such as whether the system will be more likely to collapse when there is a certain downward pressure from the palm on the CRS system? Stability while using this system can be an interesting discussion aspect, for the authors' information.

Authors’ reply: The CRS does reduce the load-bearing capacity of the haptic system. The reason lies in that the CRS must not collapse, in addition to the condition of a haptic

system that the actuator output force should be sufficiently large. In Fig. 3a-d, we provide a mechanical model of the CRS and the no-collapse condition Eq.(4). For the case pointed out by the reviewer, once the elastic foundation coefficients corresponding to the palms of the hands are known, the parameters such as the thickness and width required for the CRS not to collapse can be calculated.